# Local enrichment of HP1alpha at telomeres alters their structure and regulation of telomere protection

Tracy T. Chow [1], Xiaoyu Shi [2], Jen-Hsuan Wei[1,3], Juan Guan[2], Guido Stadler[4], Bo Huang [2,5] & Elizabeth H. Blackburn [1,6]

Enhanced telomere maintenance is evident in malignant cancers. While telomeres are thought to be inherently heterochromatic, detailed mechanisms of how epigenetic modifications impact telomere protection and structures are largely unknown in human cancers. Here we develop a molecular tethering approach to experimentally enrich heterochromatin protein HP1α specifically at telomeres. This results in increased deposition of H3K9me3 at cancer cell telomeres. Telomere extension by telomerase is attenuated, and damage-induced foci at telomeres are reduced, indicating augmentation of telomere stability. Super-resolution STORM imaging shows an unexpected increase in irregularity of telomeric structure. Telomere-tethered chromo shadow domain (CSD) mutant I165A of HP1α abrogates both the inhibition of telomere extension and the irregularity of telomeric structure, suggesting the involvement of at least one HP1α-ligand in mediating these effects. This work presents an approach to specifically manipulate the epigenetic status locally at telomeres to uncover insights into molecular mechanisms underlying telomere structural dynamics.

[1] Department of Biochemistry and Biophysics, University of California, San Francisco, San Francisco, CA 94143, USA. [2] Department of Pharmaceutical Chemistry, University of California, San Francisco, San Francisco, CA 94143, USA. [3] Howard Hughes Medical Institute, University of California, San Francisco, San Francisco, CA 94143, USA. [4] Berkeley Lights Inc, Emeryville, CA 94608, USA. [5] Chan Zuckerberg Biohub, San Francisco, CA 94158, USA. [6] Salk Institute for Biological Studies, La Jolla, CA 92037, USA. Correspondence and requests for materials should be addressed to E.H.B. (email: Elizabeth.Blackburn@ucsf.edu)

Telomere maintenance is indispensable for indefinite proliferation of cancer cells. Mammalian telomeres consist of tracts of hexameric DNA repeats (5′-TTAGGG-3′) bound by protective nonhistone proteins in a complex called shelterin[1,2]. Paradoxically, in spite of the nucleosome-disfavoring properties of telomeric repeats[3], mammalian telomeric DNA is also organized into closely packed nucleosomes[4]. It is unknown how the resulting telomeric chromatin domain, consisting of the telomere nucleosomal chromatin plus shelterin complex, establishes a capping structure to maintain genome integrity[5,6]. While functions associated with shelterin itself have been widely studied, molecular details of how this peculiar telomere chromatin impacts mammalian telomere maintenance remain largely unexplored.

Telomere chromatin is thought to be inherently condensed heterochromatin primarily based on findings in yeast[7,8], Drosophila[9], and mouse[10]. In these organisms, establishment of telomeric and subtelomeric heterochromatin is crucial for chromosomal end protection[5]. However, recent studies suggest that human and Arabidopsis telomere chromatins are relatively dynamic, characterized by a mix of heterochromatic and euchromatic marks, as well as enrichments of histone modifications associated with active transcription[11–14]. Besides canonical telomere capping, telomeric chromatin also regulates telomere position effect (TPE)[15], telomere transcription[16], homologous recombination at telomeres[17,18], cellular differentiation[19], and nuclear reprogramming[20].

Roles for epigenetic regulation of telomere maintenance have been sought in many studies. Knockout of various histone modifying enzymes such as histone methyltransferases SUV39H1/2, SUV4-20H1/2[10,17,21] result in defective telomere function, aberrantly increased telomere length, and chromosomal instability. Depletion of yeast histone methyltransferase Dot1[22] and its homolog in mouse (Dot1L)[23], mammalian histone modifier ATRX and its chaperon DAXX[24,25], yeast histone deacetylases Sir2[26] and its orthologs in mouse (Sirt1)[27] and human (Sirt6)[28] result in a range of altered or defective telomere maintenance phenotypes. These include alterations in telomere length[10,21], recombination which characterizes alternative telomere lengthening[10,17,29], TPE[15], telomere transcription[25], DNA damage at the telomeres[27], or increased telomere fusion and premature senescence[28]. However, in such knockout or knockdown studies, it is very difficult to interpret the molecular mechanisms underlying the dynamics of telomeric chromatin because they take place in settings of global genomic changes in chromatin and histone modifying enzymes. Therefore, we desired to set up an alternative approach to engineer localized manipulations of telomere chromatin.

A common feature of heterochromatin-mediated telomere protection in Drosophila and yeast is that their telomeric and subtelomeric chromatins respectively are enriched in heterochromatin marks such as trimethylation of lysine 9 of histone H3 (H3K9me3)[30]. H3K9me3 provides a high affinity binding site for HP1 (heterochromatin protein 1), and recruits histone methyltransferase SUV39H to catalyze the propagation of this mark to establish heterochromatin[31]. Extensive studies of heterochromatin marks, using chromatin immunoprecipitation (ChIP) and genome-wide chromatin state mapping, have reported enrichment of H3K9me3 and other heterochromatin marks in mouse subtelomere and telomeres[30]. In striking contrast to this reported high H3K9me3 at mouse telomeres, unexpectedly low density of telomere H3K9me3 and rather infrequent HP1 are naturally localized at human telomeres[11,14,32–35]. This provides an opportunity to enhance the presence of this naturally occurring component of telomeric chromatin to study its role in telomere biology.

In this report, we present an approach to study the consequences of locally altering telomere chromatin properties on the key functions of telomeres. We enrich heterochromatinization at telomeres by fusing HP1alpha (HP1α) to the telomere binding shelterin protein TRF1. We find that deposition of heterochromatin marks at telomeres is increased and telomerase-mediated telomere extension is attenuated. Mutational studies of such telomere-tethered HP1α show the chromo shadow domain (CSD) of the telomere-tethered HP1α is involved in attenuating telomere extension. Additionally, DNA-damage responses at telomeres, triggered by either expressing mutant-template telomerase RNA (hTR) or depletion of shelterin TRF2, are reduced, suggesting enhanced telomere stability. Direct super-resolution visualization of this HP1α-tethered telomere chromatin in cells by stochastic optical reconstruction microscopy (STORM) imaging shows previously unsuspected less globular, more irregularly shaped telomere structures. These findings provide a platform for understanding the crosstalk between altered chromatin environment, epigenetic regulation and telomere maintenance.

## Results

**A model system to study HP1α function at telomeres**. To study how altered telomere chromatin regulates its maintenance, we set up a controlled system to enhance heterochromatin in a locus-specific manner. We fused shelterin TRF1, which confers telomeric locus-specificity, to HP1α, a protein involved in heterochromatin establishment and maintenance. HP1α contains a conserved N-terminal chromo domain (CD) that binds to dimethylated and trimethylated H3K9 (H3K9me2/3) and a C-terminal CSD for dimerization and ligand binding[31,36]. These two domains are joined by a flexible hinge domain (Fig. 1a)[31].

To validate our system, EGFP-tagged TRF1 fused with HP1α (Fig. 1a) was transiently cotransfected with mCherry-tagged TRF2, a core shelterin component, and tested for colocalization at telomeres (Fig. 1b) in human bladder cancer UM-UC3 cells. As expected, EGFP-HP1α is capable of localizing to nontelomeric genomic regions, resulting in a significantly higher total average HP1α occupancy (~16.7% area per nucleus) compared to EGFP-TRF1 (~5.0%) that localized exclusively to telomeres (Fig. 1c), as measured by percent EGFP per nucleus. Meanwhile, TRF1HP1α also localized to genomic regions other than telomeres with no significant difference of average nucleus occupancy (~17.3%) compared to control HP1α (~16.7%). Thus, TRF1HP1α also retained the functional abilities of HP1α for targeting and chromatin spreading (Fig. 1c). A point mutation in the CD domain of the TRF1HP1α-fusion construct (V22M), which abrogates recognition of H3K9me3 by HP1α, maintained its ability to localize at telomeres, as will be discussed further below, and reverted average EGFP occupancy in the nucleus to ~6.4%. Average colocalization with TRF2 was significantly higher for both EGFP-TRF1HP1α (~74.2%) and EGFP-TRF1 (~62.1%) compared to EGFP-HP1α alone (~46.9%) (Fig. 1d). Thus, TRF1HP1α is expressed and specifically enriched at telomeres.

**TRF1HP1α expression increases H3K9me3 per H3 at telomeres**. In addition to microscopy, we also used ChIP to follow the genomic localization of stably expressed TRF1HP1α cells (Fig. 1e–j). Immunoprecipitated chromatin was hybridized with either telomeric or control centromeric (CENPB) probe (Fig. 1f). After normalizing to intensity of 10% total chromatin input, TRF1HP1α showed ~28-fold increased average HP1α at telomeres compared to controls (Fig. 1f, g). While TRF1 overexpression resulted in a slight decrease of H3 at telomeres compared to vector only (Vonly) or HP1α, each of the three

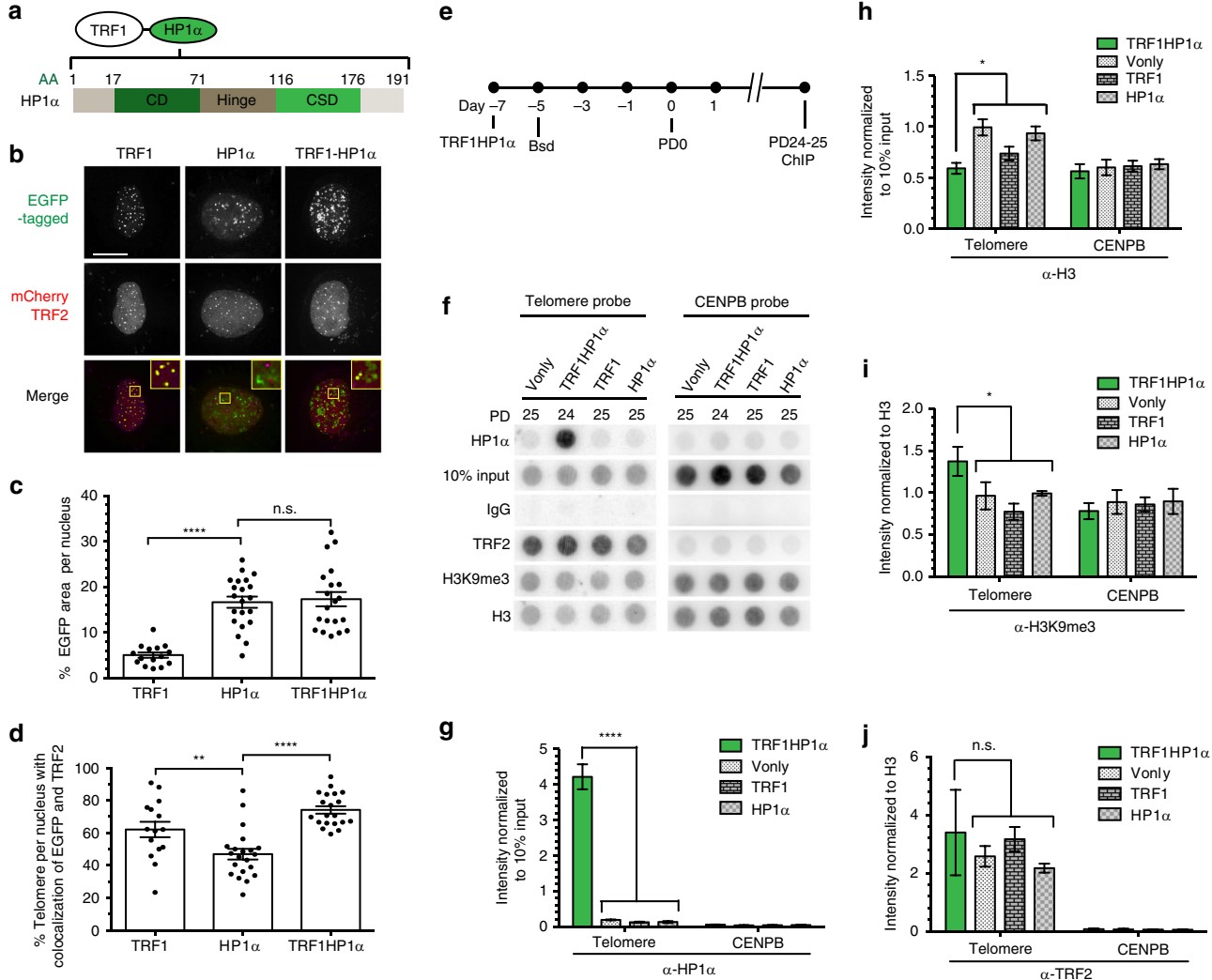

**Fig. 1** Tethered HP1α at telomeres locally increases H3K9me3. **a** Schematic of HP1α fused to TRF1. HP1α consists of a chromo domain (CD), a hinge, and a chromo shadow domain (CSD); AA (amino acid). **b**−**d** Fluorescence imaging of UM-UC3 cells cotransfected with mCherry-tagged TRF2 and EGFP-TRF1, EGFP-HP1α, or EGFP-TRF1HP1α-fusion 48 h after transfection (*n* = 15–21 nuclei). **b** Representative images. mCherry shown as magenta in merged image. Scale bar: 10 μm. **c** Quantification of % EGFP area per nucleus ****p < 0.0001; n.s. (no significance). **d** Quantification of % telomere per nucleus with colocalization of EGFP and TRF2 (mCherry) **p = 0.0065; ****p < 0.0001. The high apparent colocalization of HP1α with TRF2 (within the HP1α group) is partly caused by random, coincidental overlaps with telomeres due to widespread HP1α spots; X−Y planes are projections of z-stacks. **c**, **d** Significance is assessed by one-way ANOVA and Dunnett's multiple comparison test with 95% confidence level. Error bars represent standard error of the mean (s.e.m.). **e** Experimental set-up for ChIP to follow the localization of stably expressed TRF1HP1α in UM-UC3 after blasticidin selection (Bsd) at ~PD25. **f** Experimental groups are immunoprecipitated with the indicated antibodies, and hybridized on a dot blot with either telomere or control centromere (CENPB) probe (*n* = 3 independent replicates). Upon signal normalization to 10% input, **g** TRF1HP1α shows increased HP1α at telomeres compared to controls Vector only (Vonly), TRF1 and HP1α ****p < 0.0001; **h** TRF1HP1α shows decreased H3 at telomeres *p = 0.0133. Upon normalization to H3 signal, **i** TRF1HP1α shows increased H3K9me3 at telomeres per H3 *p = 0.0101 while **j** there is no significant change of TRF2 occupancy at telomeres. n.s. (no significance) **g**−**j** The values for three independent experiments (Supplementary Fig. 1) are used to calculate the s.e.m. for each group. p values are calculated by two-tailed unpaired t test with 95% confidence level

control groups showed higher H3 compared to TRF1HP1α (Fig. 1f, h). Combining all three control groups, TRF1HP1α showed less H3 (~0.7 fold) per telomere (Fig. 1f, h). We then asked if H3K9me3 heterochromatin marks at telomeres were increased. Upon normalizing to telomeric H3, TRF1HP1α showed a small but significant (~1.5-fold) increase of H3K9me3 at telomeres. (Fig. 1f, i). Meanwhile, there was no significant change in TRF2 occupancy, a core component of shelterin complex (Fig. 1f, j). Moreover, TRF1HP1α by itself did not induce DNA damage at telomeres, as will be discussed in detail below, suggesting shelterin integrity remained intact. See Supplementary Fig. 1 for independent, uncropped images of triplicate ChIP

experiments. In summary, we established a controlled system to alter telomere heterochromatin by HP1α tethering, resulting in increased H3K9me3 at telomeres.

**TRF1HP1α attenuates telomere extension**. To investigate if tethered HP1α-induced heterochromatin regulates telomere extension by telomerase, EGFP-tagged TRF1HP1α or corresponding control groups (Vonly, TRF1, HP1α) were introduced into UM-UC3 cells via lentiviral construct infection. Blasticidin-selected cells were FACS sorted for medium EGFP expression (assigned as Population Doubling PD0). Protein expression was

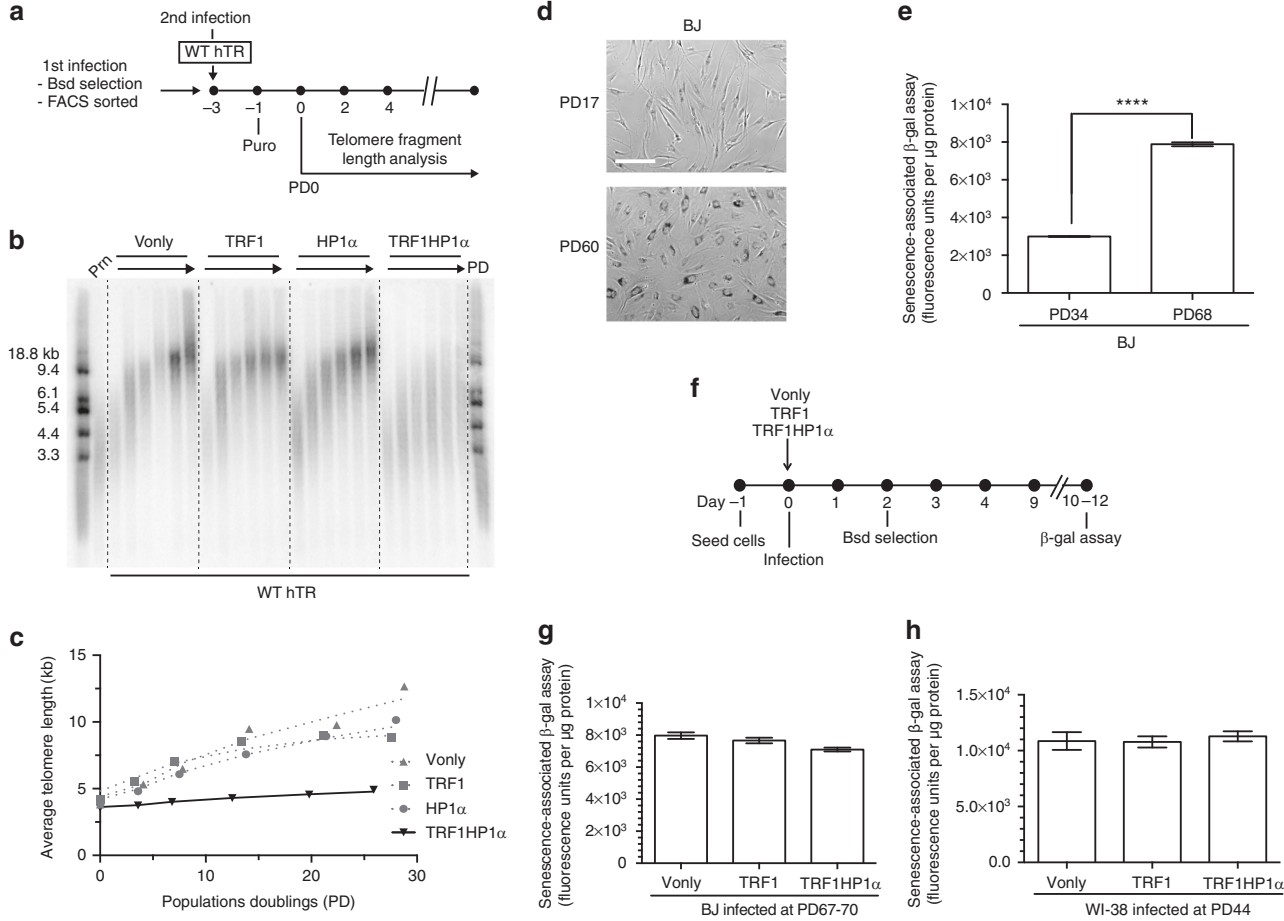

**Fig. 2** Telomere-tethered HP1α attenuates telomere extension by telomerase but does not accelerate replicative senescence. **a** Experimental set-up to study the impact of HP1α on telomerase-based telomere extension in UM-UC3. First infection: EGFP-tagged Vonly, TRF1, HP1α, or TRF1HP1α. **b** Telomere length analysis of TRF1HP1α, various controls, and untreated parental cells (Prn) with WT hTR overexpression from PD0 to ~PD30. **c** Quantification (average telomere length) shows TRF1HP1α attenuates the WT hTR overexpression-induced telomere extension. Similar findings are observed in two independent replicates. **d** Qualitative β-gal staining of BJ fibroblasts with earlier versus later PD. Bar: 100 μm. **e** Quantifications of relative β-gal fluorescence units are normalized to μg of protein. BJ PD68 shows significantly more β-gal fluorescence than BJ PD34 ****$p < 0.0001$. Two independent experiments; each contains triplicates. Error bars represent s.e.m. $p$ values are calculated by two-tailed unpaired $t$ test with 95% confidence level. **f** Experimental set-up to determine if TRF1HP1α accelerates replicative senescence. These analyses were performed only 10–12 days after infection, and during that period (~5–6 PDs) telomere shortening was minimal. Thus, it is unlikely that the lack of any effect on β-gal was due to adaptive compensation by other proteins or selection of cell subpopulations. Fibroblasts **g** BJ (PD67–70) or **h** WI-38 (PD44) show no significant difference in β-gal signal. BJ, three independent experiments each contain triplicates. WI-38, single experiment with triple replicates. Error bars represent s.e.m.

validated by western blot analysis (Supplementary Fig. 2). All overexpression cell lines showed only minimal alteration in telomere length up to ~PD80 (Supplementary Fig. 3). This observation is consistent with a previous report that only long-term culturing of TRF1 overexpression in certain cancer cells resulted in telomere shortening[37]. To better resolve changes in length, telomere extension was enhanced by overexpressing WT hTR (template specifying 5′-TTAGGG-3′ repeats), which we have previously shown lengthens telomeres in UM-UC3 cells during the following few days in culture[38]. WT hTR was introduced via a second round of infection with the experimental set-up diagrammed in Fig. 2a. Southern blotting (Telomere Restriction Fragment Length) analysis showed that the telomere-tethered TRF1HP1α expression attenuated telomere extension compared to Vonly, TRF1-alone, or HP1α-alone controls (Fig. 2b, c).

Uncapped telomeres elicit senescence in cultured human fibroblasts. We used the senescence-associated beta-galactosidase (β-gal) assay to determine if TRF1HP1α influenced replicative senescence. High PD normal human foreskin fibroblast BJ cells showed the expected increase of β-gal fluorescence units (~2.6-

fold higher than at lower PD; Fig. 2d, e). However, in two primary fibroblast cell lines, BJ or WI-38, there were no significant differences among TRF1HP1α or corresponding Vonly or TRF1 control groups (Fig. 2f–h). Thus, tethered HP1α at telomeres did not exacerbate replicative senescence in fibroblasts, further validating the intact functionality of the manipulated telomeric chromatin domain.

**Tethering TRF1HP1α containing mutations within HP1α.** To rule out potential indirect effects due to tethering of TRF1HP1α to nontelomeric HP1α genomic loci and to understand mechanistically how HP1α inhibited telomere elongation, HP1α constructs carrying various characterized separation-of-function mutations fused with TRF1, as above, were introduced into UM-UC3 cells (Fig. 3a): (i) CD mutant V22M[39], defective in recognizing H3K9me3 marks; CSD mutants (ii) I165A[39], deficient in dimerization and ligand binding and (iii) W174A[39], which can dimerize but is deficient in ligand binding; (iv) N-terminal phosphorylation mutant NS2A[40], to perturb oligomerization; and

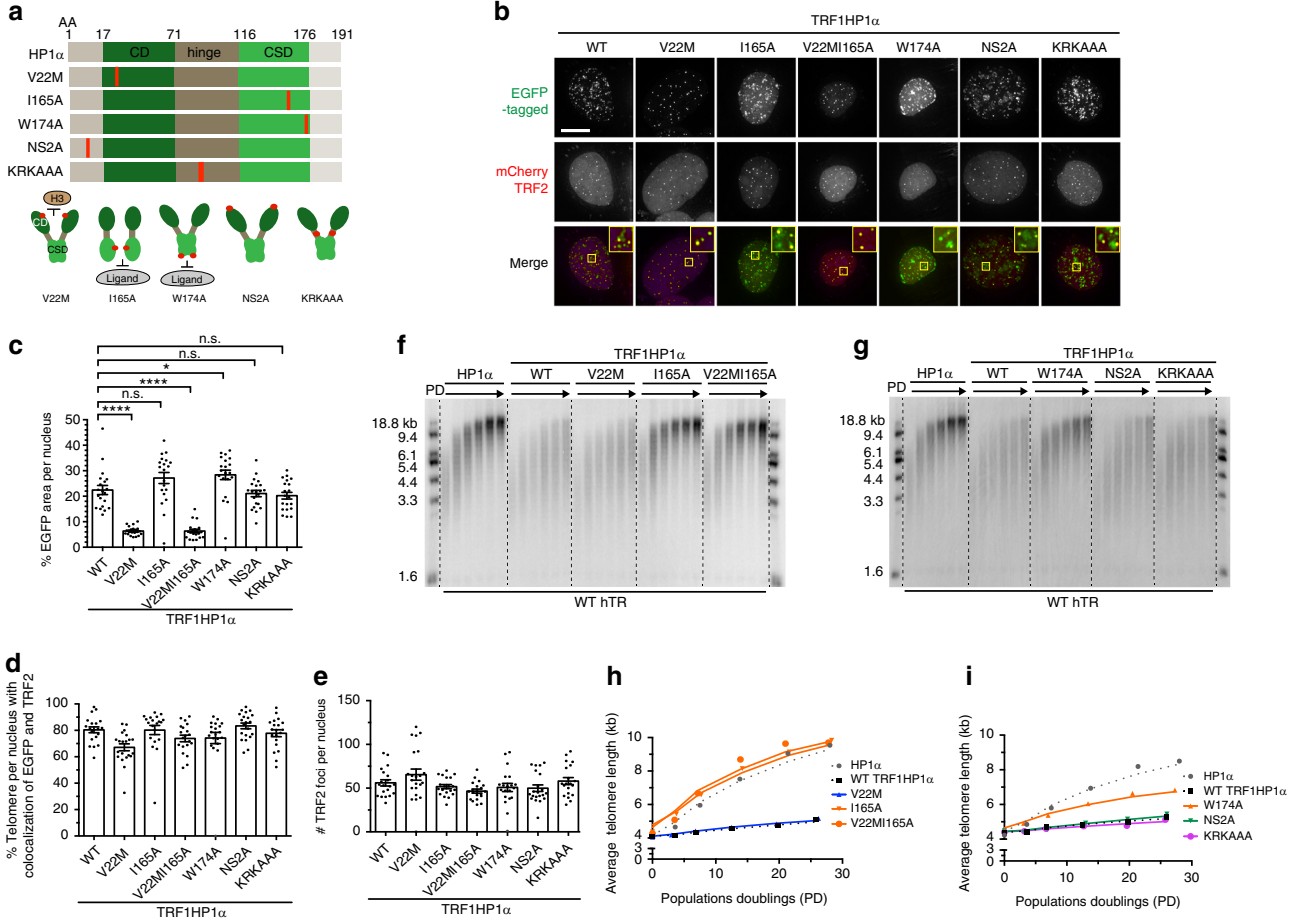

**Fig. 3** Ligand binding function of HP1α CSD controls telomere extension. **a** Schematic diagram of mutations in HP1α fused to TRF1 (AA—amino acid). CD mutant V22M; CSD mutants I165A and W174A; N-terminal phosphorylation deficient mutant NS2A; hinge mutant KRKAAA. **b** Transient cotransfection of mCherry-tagged TRF2 (magenta in merged image) and various EGFP-tagged TRF1HP1α mutants respectively in UM-UC3 cells imaged after 48 h. Scale bar: 10 μm. ~20 nuclei were counted per group in **c** and **d**. **c** Quantification of % EGFP area per nucleus ****$p < 0.0001$; *$p = 0.0260$; n.s. (no significance). Significance is assessed by one-way ANOVA and Dunnett's multiple comparison test with 95% confidence level. **d** Quantification of % telomeres per nucleus with colocalization of EGFP and TRF2 (mCherry). Consistently, V22M and V22MI165A show fewer total fusion protein spots per nucleus (Supplementary Fig. 4) because V22M lacks the ability to bind to other, widespread genomic regions. Thus, the slight reduction of % colocalization of V22M and V22MI165A with TRF2 is likely to be at least partially because of fewer random overlaps of telomeres with widespread HP1α spots. **e** Quantification of TRF2 foci; $n = $ ~20 nuclei per group. **c**−**e** Error bars represent s.e.m. **f**, **g** Telomere length analyses of TRF1HP1α, WT or HP1α mutant variants with WT hTR overexpression across PD0 to ~PD30. **h**, **i** Quantifications (average telomere length) show CSD mutants I165A, W174A or double mutant V22MI165A revert the telomere extension attenuation phenotype of TRF1HP1α. Similar findings were observed in two independent experiments

(v) hinge mutant KRKAAA[36,41], deficient in HP1α DNA / RNA interaction (Fig. 3a).

Validation of the ability of these mutant proteins to localize to telomeres or other genomic regions was performed as described for Fig. 1b–d. WT TRF1HP1α and all mutants tested had considerable amounts of tethering to other genomic regions except for V22M or V22MI165A (which do not recognize H3K9me3) (Fig. 3b). Average HP1α nucleus occupancy was reduced in V22M (~6.4%) and the double mutant V22MI165A (~6.3%), but not I165A (~27.1%), compared to WT TRF1HP1α (~22.5%) (Fig. 3b, c). Consistent patterns were observed by quantifying total numbers of fusion protein spots per nucleus (Supplementary Fig. 4). Thus, loss of H3K9me3 binding by V22M or V22MI165A resulted in deficient anchorage to nontelomeric chromatin.

However, all mutants, including V22M and V22MI165A, were efficiently tethered at the telomeres via their fused TRF1 (~67.1–83.4% colocalization; Fig. 3b, d). Thus, in this controlled tethering system, telomere anchorage of V22M was efficiently driven by its TRF1 fusion and did not require HP1α recognition

of H3K9me2/3, that might potentially have contributed to nontelomeric localization. Therefore, we deliberately used V22M to control for possible indirect effects due to tethering of TRF1 to nontelomeric HP1α genomic sites. Meanwhile, there was no significant change in number of TRF2 foci per nucleus (Fig. 3e).

**Chromo shadow domain of HP1α attenuates telomere extension.** To determine which domain functions of HP1α control telomere extension by telomerase, we generated cells stably overexpressing TRF1HP1α-constructs harboring various mutations within HP1α (Fig. 3a–d), using the experimental set-up shown (Fig. 2a). Interestingly, WT TRF1HP1α and V22M limited telomere extension to similar extents (Fig. 3f, h). Hence, because TRF1 tethering of HP1α to telomeres bypassed the need for H3K9me2/3 recognition for HP1α recruitment to telomeres, HP1α recognition of H3K9me2/3 per se was not required for this inhibition of telomere extension. In contrast, I165A abolished the inhibition of telomere lengthening, as did V22MI165A (Fig. 3f, h). Since I165A abrogates both dimerization and ligand

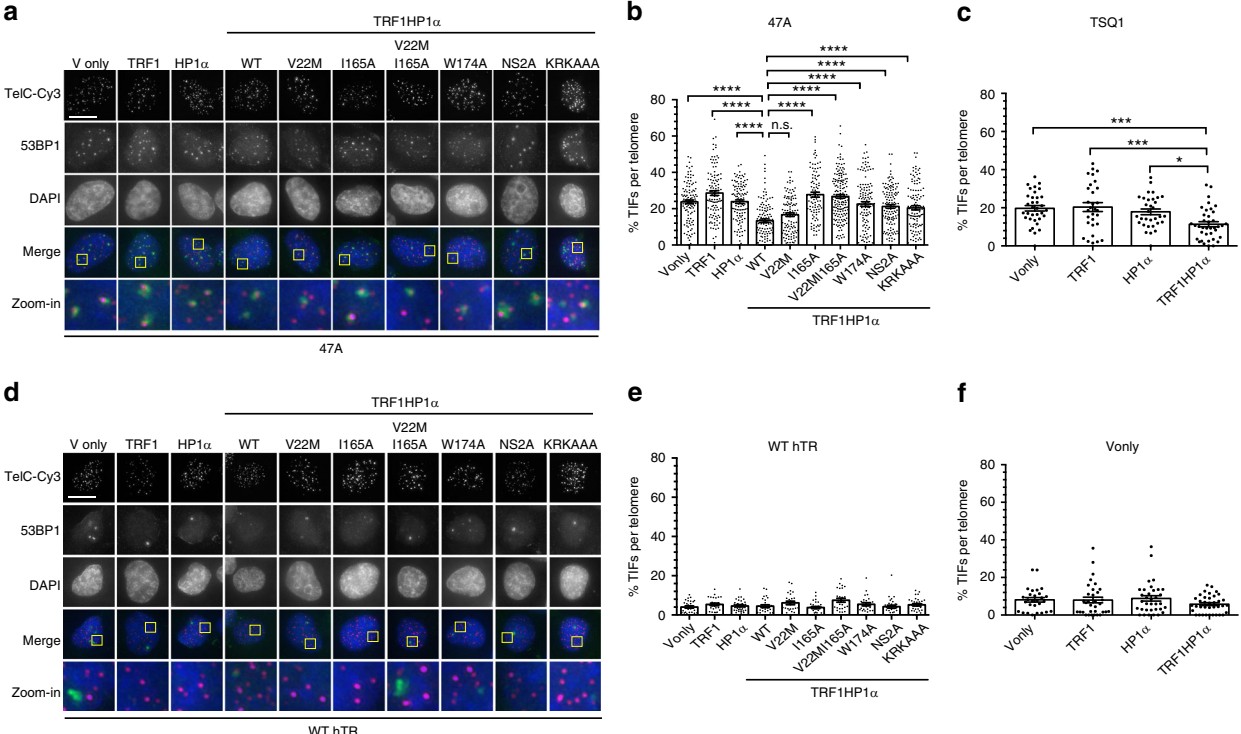

**Fig. 4** TRF1HP1α results in reduced TIFs induced by mutant hTR expression. Cells stably expressing TRF1HP1α (WT or mutant variants of HP1α) were infected with lentivirus containing WT hTR, or mutant hTR (47A or TSQ1) on day 0, selected for stable expression after 48 h, and analyzed on day 5. **a** Fluorescence microscopy images of representative cells expressing mutant hTR 47A stained with telomeric (Tel-Cy3) peptide nucleic acid (PNA) probes (magenta in merged image) via fluorescent in situ hybridization (FISH), antibody against DNA-damage repair protein marker 53BP1 (green in merged image), and counterstained with DAPI. Zoom-in images (the last row) correspond to yellow-squared regions of the row above. **b** % TIF per telomere of each nucleus is quantified; $n = 94–159$ nuclei combining data of three independent experiments. ****$p < 0.0001$; n.s. (no significance). **a**, **b** Scale bar: 10 μm. **c** Upon TSQ1 expression, TRF1HP1α results in fewer TIFs compared to Vonly, TRF1 or HP1α controls. TRFHP1α ~11.4% shows decreased TIFs compared to Vonly: ~19.7% ***$p = 0.0008$; TRF1: ~20.4% ***$p = 0.0005$; HP1α: ~17.9% *$p = 0.0137$ ($n = 30–38$ nuclei). **b**, **c** Significance is assessed by one-way ANOVA and Dunnett's multiple comparison test with 95% confidence level. **d** Fluorescence images of control cells overexpressing WT hTR ($n = 27–36$ nuclei). Same color scheme as **a**. TIFs quantification in the presence of **e** WT hTR or **f** Vonly ($n = 28–36$ nuclei) show minimal baseline DNA damage at telomeres. **b**, **c**; **e**, **f** Error bars represent s.e.m.

binding, we sought to separate which function was primary in this regulation of telomerase action. An additional CSD mutant W174A, which is deficient in ligand binding but can still dimerize, only partially restored the inhibition of lengthening rate (Fig. 3g, i). Thus, because dimerization was not sufficient to fully inhibit telomerase action down to the WT TRF1HP1α level, the ligand binding (and possibly also dimerization) function of CSD is required to inhibit telomere extension. Finally, N-terminal phosphorylation and the hinge DNA-binding domain were not required to inhibit telomere extension (mutants NS2A and KRKAAA in Fig. 3g, i).

**TRF1HP1α reduces telomere damage induced by mutant hTR.** Knowing that TRF1HP1α inhibited telomere extension (Figs. 2, 3), using an independent readout for telomerase function, we determined whether TRF1HP1α-induced inhibition of telomerase would lead to less incorporation of mutant hTR-specified telomeric DNA, and hence lead to a diminished DNA-damage response at telomeres. Incorporated mutant telomere repeats cannot bind shelterin proteins, and lead to rapid uncapping and localized telomere damage foci[42]. Cells were infected on day 0 with WT hTR or mutant hTRs, either 47A (5′-TTTGGG-3′)[38] or TSQ1 (5′-GTTGCG-3′)[43] and selected for stable expression after 48 h. On day 5, 53BP1 DNA-damage foci present at telomeres, also referred to as telomere dysfunction-

induced foci (TIFs), were increased (Fig. 4a–c) compared to WT hTR (Fig. 4d, e). We tested TIF induction early, when cell growth was only mildly affected (Supplementary Fig. 5). Introduction of TRF1HP1α yielded fewer average 47A-induced TIFs (~13.4%) compared to controls Vonly (~23.8%), TRF1 (~28.6%) and HP1α (~23.8%) (Fig. 4a, b). Similar findings were also observed with TSQ1 treatment (Fig. 4c). Moreover, WT TRF1HP1α (~13.4%) and V22M (~16.7%) showed similar TIFs (Fig. 4a, b). However, elevated TIFs were observed in CSD mutants I165A (~27.7%), W174A (~22.6%), and V22MI165A (~26.8%). In cells over-expressing WT hTR, minimal baseline DNA damage at telomeres was observed in corresponding controls (ranging from 4.1–7.5%; Fig. 4d, e) or Vonly (5.8–8.9%; Fig. 4f).

In these experiments, the DNA damage caused by incorporated mutant repeats depends on telomerase action at telomeres. We showed that WT TRF1HP1α inhibited telomere extension to similar extents as mutants V22M, NS2A, and KRKAAA (Fig. 3). If reduced TIF levels were solely due to telomerase inhibition, we would expect that TIF induction upon 47A expression would be similar with all four fusion proteins. However, notably, upon 47A expression, NS2A and KRKAAA showed more TIFs compared to WT and V22M TRF1HP1α (Fig. 4a, b). These results indicate a separation of HP1α functions: on the one hand, in regulating telomere extension via its C-terminal CSD (ligand binding and dimerization) and on the other hand, in DNA-damage reduction (via its N-terminal CD and hinge domains).

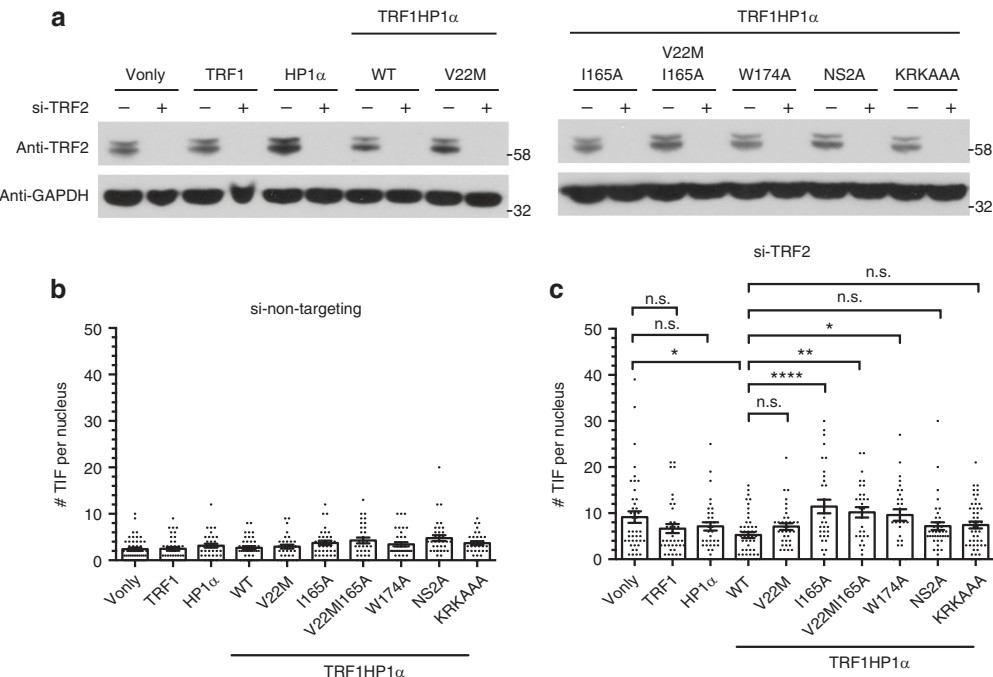

**Fig. 5** TRF1HP1α allele-specific protection effects upon si-TRF2-induced telomeric damage. 72 h after transfection, **a** TRF2 knockdown efficiency with antibody against TRF2 (anti-TRF2) and GAPDH (anti-GAPDH) as loading control. (−) si-non-targeting; (+) si-TRF2. Quantification of TIFs in **b** si-non-targeting ($n = 32$–47 nuclei per group) or **c** si-TRF2. Left *$p = 0.0188$, right *$p = 0.0192$, **$p = 0.0042$, ****$p < 0.0001$ ($n = 31$–48 nuclei per group). **b**, **c** Significance is assessed by one-way ANOVA and Dunnett's multiple comparison test with 95% confidence level. Error bars represent s.e.m. Note the similar pattern among TRF1HP1α alleles in **c** compared to the corresponding allele pattern in Fig. 4b

**Tethered HP1α reduces telomere damage induced by si-TRF2.** To further study the direct telomere-protective effect of HP1α, we used two additional, independent approaches. First, we induced telomere damage by efficiently knocking down TRF2 with si-TRF2 (Fig. 5a). Baseline TIFs were quantified using control non-targeting siRNA (Fig. 5b). TRF1HP1α mildly protected from si-TRF2-induced telomere damage (Fig. 5c). Furthermore, comparing across all of the TRF1HP1α mutants, the pattern of allele-specific effects on TRF2-depletion-induced TIFs closely paralleled their corresponding pattern on 47A-hTR-induced TIFs (compare Fig. 4b with Fig. 5c). This similarity of protective effects, against both telomerase-independent (TRF2 knockdown) and telomerase-dependent (47A hTR-induced) damage, indicates that in addition to its inhibitory effect on telomerase action, telomere-tethered WT TRF1HP1α can also protect telomeres.

Independently, we also developed a CRISPR/Cas9-based telomeric DNA-cutting strategy to induce telomere-specific damage in cells (Supplementary Fig. 6). Interestingly, expressing either TRF1 alone or telomere-tethered WT TRF1HP1α reduced CRISPR-induced telomere DNA cutting to similar extent in UM-UC3 cells. In summary, employing different approaches to induce telomeric damage has uncovered different aspects of how tethered HP1α affects telomere protection.

**TRF1HP1α increases irregularly shaped telomere structures.** Telomere structures are smaller than the diffraction-limited resolution (~250 nm) of conventional light microscopy[44–46]. Under stochastic optical reconstruction microscopy (STORM), the great majority of WT telomeres appear as spherical, globular structures[44–46]. Using STORM, we examined whether HP1α tethering altered the size or globular shape of telomeres. Under our conditions, three-dimensional (3D) STORM provided XY precision of ~30 nm and Z resolution of ~70 nm[47]. Cells stably expressing TRF1HP1α, or corresponding control groups (TRF1,

HP1α), were collected for telomere length analysis or fixed for STORM analysis. We first verified that all experimental groups, collected at earliest passage after blasticidin selection (days 8–9 post lentiviral infection), showed similar population telomere lengths (Fig. 6a). Therefore, any observed telomere shape changes at the population level should not be a result of average telomere length alteration.

3D STORM showed significantly better resolution compared to conventional widefield imaging (Fig. 6b, top and middle panels). The overlay image also allowed us to exclude any nontelomeric background, ensuring the identified clusters correspond to telomeres (Fig. 6b, bottom panel). To quantify structural changes of individual telomeres, we measured the radius of gyration (Rg) of each cluster. Rg represented the root-mean-square distance of the localization points from the center of mass of a cluster according to $R_g^2 = (1/N) \sum_{k=1}^{N} (\vec{r}_k - \vec{r}_{center-of-mass})^2$, where $\vec{r}$ denotes position, $k$ denotes the localization point index, and $N$ is the number of localization points. The average number of localization points of such filtered individual telomeres for TRF1, HP1α, WT TRF1HP1α, TRF1HP1αI165A were 664, 420, 544, and 639, respectively (Supplementary Fig. 7). As an imaging quality control, we only analyzed telomere clusters with centers of mass near the focal plane, and consisting of more than 200 localization points (Fig. 6c, bottom panel). Telomeric localization points were clustered using Insight3 software[47] to reconstruct structures of individual telomeric foci (Fig. 6c, top panel). Across all experimental groups, individual Rg values showed only weak correlations with number of localization points (Supplementary Fig. 7). Average Rg was similar in parental cells and Vonly, suggesting any observable changes in Rg were not caused by the vector itself (Supplementary Fig. 8).

Some generalities emerged from these analyses. As expected, most telomeres appeared spherical, but heterogeneous shapes

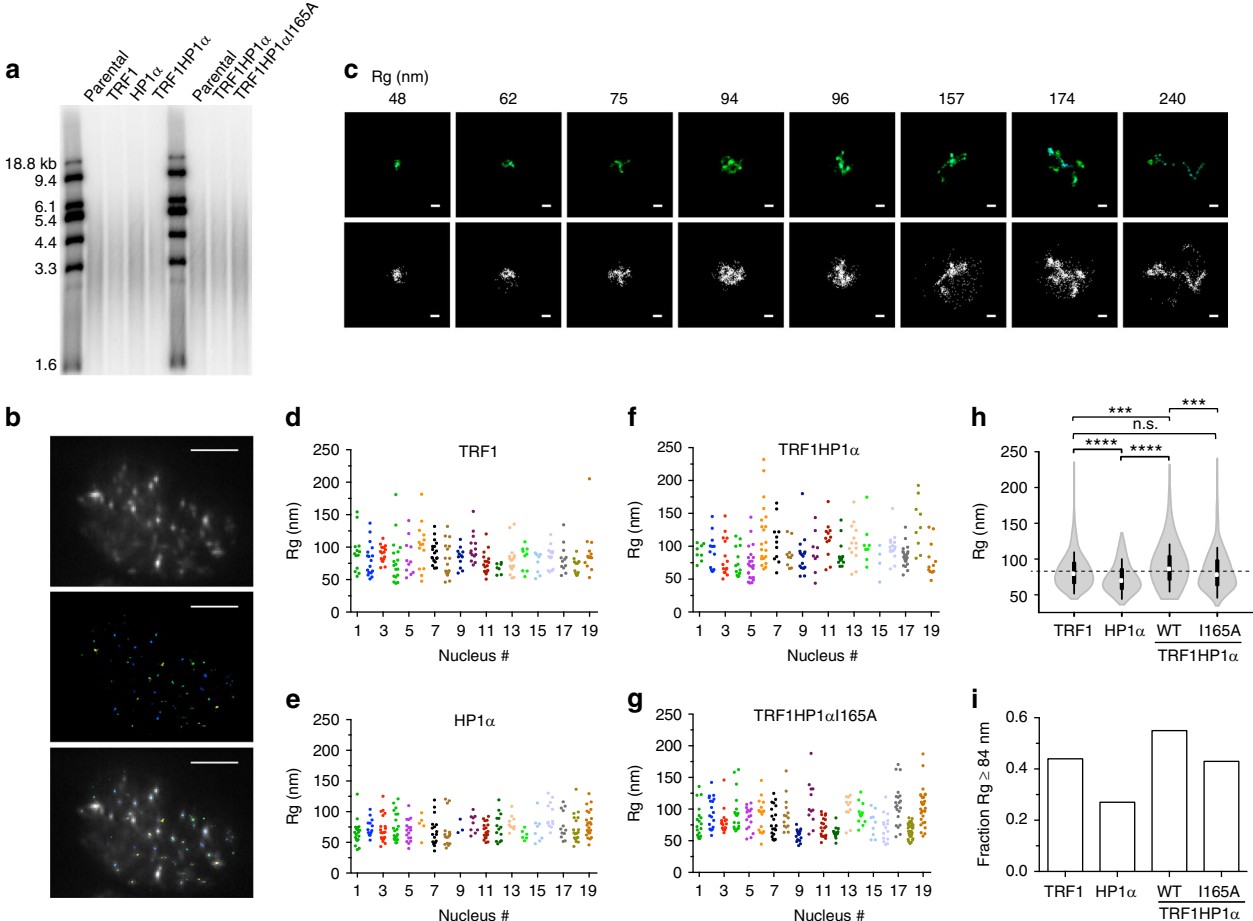

**Fig. 6** TRF1HP1α increases the fraction of irregular-shaped telomeres analyzed by STORM. **a** Similar telomere length (average and length distribution) across TRF1HP1α, I165A and control groups (TRF1, HP1α) at the time of analysis. **b** Top: Widefield conventional fluorescence image of representative UM-UC3 nucleus hybridized with Cy5-end-labeled C-strand telomeric PNA FISH probe. Images acquired contain ~35,000 frames with a z-depth-range of ~700 nm. Middle: the corresponding STORM image. Bottom: Overlay of conventional and STORM images. Bar: 5 μm. **c** Top: Representative reconstructed single telomere STORM images of TRF1HP1α and each corresponding Rg (nm) across a gradient. Bottom: Corresponding raw images of individual signal localization spots (displayed as dots) prior to image processing and reconstruction. Bar: 100 nm. **d–g** Rg of individual telomeres (dots) in 19 nuclei analyzed for each group **d** TRF1, **e** HP1α, **f** TRF1HP1α or **g** TRF1HP1αI165A. Y-axis: Rg (nm). X-axis: nucleus index. Each individual nucleus is distinguished by a different color. Each dot corresponds to one telomere. **h** Distribution of Rg (nm) represents as a violin plot showing frequency (width of density plot), median (white dot), interquartile range (bar), and 95% confidence interval (line). TRF1 ($n = 38$ nuclei, 437 telomeres), HP1α ($n = 19$ nuclei, 264 telomeres), TRF1HP1α ($n = 47$ nuclei, 552 telomeres), and TRF1HP1αI165A ($n = 27$ nuclei, 451 telomeres). Means of Rg are compared using ANOVA Tukey's multiple comparisons with 95% confidence level ****$p < 0.0001$; left ***$p = 0.0003$; right ***$p = 0.0001$; n.s. (no significance). Mean of TRF1 Rg (84 nm) indicated as cut-off (dashed line) and **i** fractions of Rg equal or greater than the 84 nm cut-off in experimental groups

were also observed[45]. Figure 6c showed examples of individual telomere structures across a gradient of Rg in TRF1HP1α. Analyses showed telomeres with larger Rg displayed more variable and irregular shapes; specifically, while more spread out in three dimensions, they were compact (dense) in one dimension (Fig. 6c). The distributions of Rg heterogeneity among individual telomeres were consistently observed in multiple nuclei for each experimental group (Fig. 6d–g). This indicated that the observed structural differences among groups, as described below, were unlikely to have been simply skewed by specific nuclei that harbored Rg outliers.

To compare among the groups, we quantified the differences in telomeric structures. Rg distribution frequency of individual telomeres were represented by violin plots (Fig. 6h). Surprisingly, the Rg mean of WT TRF1HP1α (90.7 nm) was significantly higher than the mean Rgs of controls TRF1 (84 nm) and HP1α (73.8 nm). The phenotype of the point mutant TRF1HP1α I165A (Rg mean 83.6 nm) resembled that of the TRF1 control (84 nm). We also noted that the Rg mean of TRF1 alone versus HP1α

alone differed. Further studies are underway to better understand this phenomenon. We focused our analyses on the finding that the Rg mean of WT TRF1HP1α was significantly higher than both controls (TRF1 or HP1α) or point mutant TRF1HP1α I165A. To quantify the proportions of irregular telomere structures, mean Rg of TRF1 (84 nm) was applied as a reference cut-off (Fig. 6h, dashed line). Fractions of telomeres with Rg equal or greater than 84 nm were calculated (Fig. 6i). There was a higher fraction of irregularly shaped telomeres in WT TRF1HP1α (0.55) compared to TRF1 (0.44) or HP1α (0.27), and mutation I165A reduced this back down to 0.43, similar to in TRF1 (0.44) (Fig. 6i). Together, these data indicate that tethering WT HP1α at telomeres results in increased irregularly shaped telomeres.

## Discussion

The establishment of a dynamic telomeric chromatin is important for the structural and functional integrity of telomeres. However, how structural determinants impact telomere maintenance is largely unknown. We experimentally enhanced

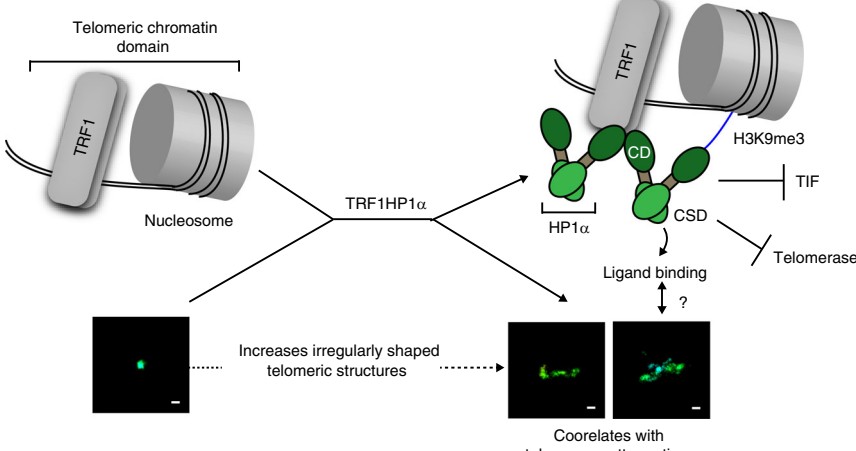

**Fig. 7** Model for how enhanced heterochromatin by telomere-tethered HP1α impacts telomere maintenance. Diagram of working model. Bar: 100 nm. Below the drawings: Representative reconstructed single telomere STORM images. See text for details

heterochromatinization at chromosomal ends by enriching HP1α specifically at telomeres. The results reported here, summarized in Fig. 7, provide insights into how heterochromatin alters telomere maintenance and structure. Using TRF1 for telomere-tethering of HP1α, which is detected naturally at telomeres but at low occupancies[14,33–35], we report that an intact dimerization domain of HP1α, with its ligand binding function, is required to regulate telomere extension. Thus, HP1α-induced chromatin alteration can function as a gatekeeper of telomerase action. The requirement for ligand binding by HP1α suggests that this function requires interaction with other factors. Moreover, employing independent modes of inducing telomere damage (mutant DNA repeat incorporation or shelterin TRF2 depletion), we find that the tethered HP1α increases telomere protection. Future studies will be of interest to determine if the telomere-localized chromatin changes induced by HP1α may also play an active role in the DNA-damage responses themselves at the telomeres. Structurally, we find that enhancing heterochromatin by tethering HP1α increases the irregularity of telomere shapes, dependent on an intact HP1α dimerization domain. This correlation suggests the possibility that certain telomeric structural conformations facilitate ligand binding efficiency to result in inhibition of telomere extension by telomerase.

Previous reports, using in vitro nucleosome reconstitution assays[3,48], suggested TRF1 and TRF2 may play roles in both the formation and dynamics of telomeric nucleosomal arrays. Telomeric DNA, like other chromosomal DNA, wraps around histone protein cores, forming nucleosomes. We observed a slight decrease of the core histone protein H3 occupancy at telomeres by overexpressing just TRF1, and a further reduction upon enriching HP1α at telomeres (Fig. 1). Decreased H3 at telomeres might reflect displacement of some nucleosomes by the tethered TRF1HP1α. This is consistent with the in vitro finding that telomere sequence disfavors nucleosome assembly[5].

TRF2, like TRF1, also directly binds double-stranded telomeric DNA[1,2]. Interestingly however, our ChIP analysis found that TRF1HP1α expression neither altered TRF2 occupancy (Fig. 1) nor elevated TIFs (Fig. 4d–f), suggesting TRF1HP1α cohabited with shelterin. We speculate that TRF1HP1α may directly interact with nucleosome-bound telomeric DNA in addition to nucleosomal-free telomeric DNA without interfering with TRF2 binding. This is consistent with previous reports, using micrococcal nuclease I mapping in mouse embryonic fibroblasts, showing no evident alteration of telomeric nucleosomal organization upon depletion of TRF2 or even the whole shelterin[4,49]. If

a significant amount of bulk TRF2 had been out-competed by TRF1HP1α for telomere binding, we would have expected a phenotype resembling that of overexpression of a dominant-negative mutant (TRF2ΔBΔM)[50], which was not observed. We cannot exclude that the balance of other shelterin components could be altered. These other components, including POT1, TIN2, RAP1, and TPP1, bind to single-stranded telomeric DNA and/or function as scaffold bridging proteins. Exactly how shelterins interplay with histones to regulate telomere dynamics are important topics for future studies.

Through these studies, we uncovered and dissected some specific functions of HP1α at telomeres. Telomerase plays a crucial role in maintaining unlimited cellular proliferation in the majority of cancer cells. Telomerase activity is regulated at multiple levels including transcriptional regulation[51,52], holoenzyme biogenesis[53], trafficking and recruitment of telomerase to telomeres[54]. However, how local telomere chromatin dynamics regulate telomerase action and telomere length has been unclear. Our HP1α mutational analyses suggest that the CSD region functions as a negative regulator of telomerase action. The CSD is required for HP1α dimerization and interaction with proteins containing a conserved motif, PXVXL[55]. Candidates for such ligands include shelterin component TIN2[56], and the telomere-associated chromatin remodeler ATRX[57], which both contain PXVXL motifs. We speculate that their recruitment by HP1α (directly or via another bridging complex) may impact telomerase action, potentially through regulating telomerase recruitment to the telomere[58,59], polymerization initiation and/or processivity[60].

A main function of the CD region for HP1α is to recognize H3K9me2/3[31]. While WT TRF1HP1α enriched HP1α at telomeres, as expected some HP1α also localized to various other genome regions, presumably harboring the recognition heterochromatin marks (Fig. 3a–c). V22M mutant lacks the ability to bind to heterochromatin marks at nontelomeric genomic regions, and was exclusively tethered by TRF1 at the telomeres, and not to other regions in the genome (Fig. 3b–d). Therefore, to exclude potential confounding effects mediated via augmented binding to such regions, we exploited mutant V22M intentionally as a control, both to eliminate any tethering by TRF1HP1α of TRF1 at nontelomeric sites, and to prevent indirect effects caused by TRF1HP1α bound to genomic regions. Telomere-tethered HP1α-directed inhibition of telomere extension was independent of H3K9me2/3 recognition by the CD. Hence, H3K9m2/3 anchoring is separable from inhibition of telomere extension.

**Table 1 Summary of experimental data describing impact of WT versus mutants TRF1HP1α on telomere lengthening and TIF (via 47A or TRF2 depletion)**

|  | Baseline | TRF1HP1α | | | | | |
|---|---|---|---|---|---|---|---|
|  |  | WT | V22M | I165A | W174A | NS2A | KRKAAA |
| Telomere lengthening | +++ | + | + | +++ | ++ | + | + |
| TIF via 47A | +++ | + | + | +++ | ++ | ++ | ++ |
| TIF via si-TRF2 | +++ | + | + | +++ | ++ | + | + |

+++ (strong telomere lengthening, high number of TIF), ++ (intermediate phenotype), + (weak telomere lengthening, low number of TIF)

Here we have reported new connections between telomere structure, protection and telomerase action (Table 1 and Fig. 7). Overexpression of TRF1HP1α increased heterochromatin mark H3K9me3 on telomeres, increased telomere protection, reduced telomerase action and surprisingly induced irregular, often visually extended, telomeric structures. Previous reports have also suggested that silent chromatin was less condensed than euchromatin since subtelomeric and pericentromeric heterochromatin regions had lower protection in micrococcal nuclease assays compared to the rest of the genome[61]. Despite the prevailing assumption that highly condensed chromatin conformation is transcriptionally inert, transcription factors were found to bind to heterochromatic repeat sequences across diverse species[62,63]. Telomeres, while thought to be more heterochromatic than other genomic regions, are transcribed into telomere repeat-containing RNA (TERRA)[16] which interacts with TRF1 and TRF2 to regulate telomere length[64]. Although molecular component changes at telomeres can trigger a switch from a protected to a deprotected state[65], our observed increased irregularity of telomere shapes occur in the absence of DNA-damage responses. We propose that these changes in telomere structures can influence protection and telomerase action. It is also possible that the reduced H3 at telomeres (Fig. 1) may influence nucleosome arrangements to result in a more irregular telomere structure.

Telomere maintenance is crucial for cancer cell proliferation. Telomere homeostasis is regulated at many different levels. Telomere chromatin encompasses highly dynamic structures interconverting between different conformations. Thus, telomere chromatin states may add another layer of protection to play an important role in regulating chromosome end maintenance and protection. Chromatin states are often altered during tumorigenesis. It has become clear that, along with genomic instability, epigenetic abnormalities promote carcinogenesis. Heterochromatin-dependent, noncanonical telomere protection strategies, resembling those found in flies or yeasts, may have been selected for some human cancers. The possibility that some cancers can adapt heterochromatin changes to stabilize their telomeres will be interesting topics for future studies. Manipulating the epigenetic status at telomeres should provide new insights for the development of innovative telomere-directed, epigenetic cancer therapeutics.

## Methods

**Cell culture**. UM-UC3 (ATCC), U2OS (ATCC), BJ (ATCC), WI-38 (ATCC), and lenti-X-293T (Clontech) cells were cultured at 37 °C in 5% $CO_2$ in high glucose DMEM medium (Hyclone, Logan, UT) containing 10% fetal bovine serum (Hyclone, Logan, UT) and 1% (vol/vol) penicillin–streptomycin (Gibco). Cotransfection was performed using PolyJet reagent (SignaGen Laboratories).

**Plasmids and lentivirus**. The pHR′ lentiviral plasmids were generated using the second-generation lentiviral system provided by Dr. Didier Trono. HP1α was a gift from Dr. Tom Misteli (Addgene plasmid # 17652)[66]. N-terminal EGFP-tagged TRF1, HP1α, WT TRF1HP1α or mutants TRF1HP1α were subcloned into pHR′ respectively with HP1α (WT or various mutants) located on the C-terminus and

TRF1 in between EGFP and HP1α. HP1α mutants were gifts from Dr. Geeta Narlikar[36]. Plasmids were driven by the CMV promoter followed by an internal ribosome entry site and a blasticidin resistance gene. pHR′ mCherry-TRF2 expression lentiviral vector contained a hygromycin resistance gene. hTR expression lentiviral vectors driven by the IU1 promoter and a puromycin resistance gene driven by the CMV promoter[42,43]. WT and mutant hTR template sequences were as follows: WT—3′-CAAUCCCAAUC-5′; 47A—3′-CAAACCCAAAC-5′ and TSQ1—3′-CCAACGCCAAC-5′. SgRNA targeting telomere (5′-caccgGTTAG GGTTAGGGTTAGGGTTA) or Gal4 (5′-caccgGAACGACTAGTTAGGC GTGTA) sequences were cloned into LentiCRISPRv2, a gift from Feng Zhang (Addgene Plasmid #52961)[67]. Lentivirus was packaged in lenti-X-293T (Clontech) using PolyJet reagent (SignaGen Laboratories). Drug selection was initiated 48 h post infection with 50 μg/ml blasticidin for 5 days (ThermoFisher Scientific). For introduction of a second round of infection with either WT or mutant hTRs, cells were selected using 8 μg/ml puromycin for 1 day (ThermoFisher Scientific).

**Western blotting**. Cells were lysed [10 mM Tris-HCl pH 7.4, 150 mM NaCl, 0.5% IGEPAL CA-630, 10% glycerol, 1 mM EDTA, 1× Halt protease inhibitor cocktail (ThermoFisher Scientific), 1 mM DTT, Benzonase nuclease 50 U/ml (Novagen)]. Lysate was spun at 13,000 rpm (15 min at 4 °C). Supernatant was heated at 95 °C for 5 min. Protein concentration was measured using Precision Red protein assay reagent (Cytoskeleton, Inc.). ~40 μg lysates were separated by SDS-PAGE and transferred onto the Immobilon P PVDF membrane (EMD Millipore). The blots were then blocked for 30 min at room temperature with 5% milk in TBST (20 mM Tris pH 7.4, 150 mM NaCl, 0.05% Tween 20) and incubated for 1 h each at room temperature with primary antibodies followed by secondary horseradish peroxidase-conjugated antibodies. After washing, the blots were treated with chemiluminescent reagents (SuperSignal West Pico kit, ThermoFisher) and exposed to films. Primary antibodies used include 1:5000 rabbit anti-GFP (A11122; Invitrogen); 1:1000 rabbit anti-TRF1 (ab1423; Abcam); 1:2000 goat anti-HP1α (ab77256; Abcam); 1:2000 goat anti-TRF2 (NB110-57130); 1:1000 mouse anti-Cas9 (A-9000; Epigentek), 1:200 mouse anti-p53 (sc-126; Santa Cruz), and 1:1000 mouse anti-GAPDH (MA515738; ThermoFisher). Secondary antibodies used include 1:5000 Goat Anti-Mouse IgG-HRP (115-035-166; Jackson ImmunoResearch), 1:5000 Goat Anti-Rabbit IgG-HRP (111-035-144; Jackson ImmunoResearch), 1:5000 Donkey Anti-Goat IgG-HRP (sc2020; Santa Cruz Biotechnology). Uncropped blots are shown in Supplementary Fig. 9.

**Chromatin immunoprecipitation and dot blot assays**. 20×10⁶ cells were trypsinized and crosslinked with 1% paraformaldehyde (w/v) (ThermoFisher Scientific) at room temperature for 5 min, followed by 125 mM glycine (Sigma) for 5 min to quench the crosslinking and washed (cold 1× PBS, 1 mM PMSF). All subsequent steps were performed at 4 °C, unless noted otherwise. Cells were resuspended into ChIP lysis buffer (0.5% NP-40, 85 mM KCl, 20 mM Tris-HCl pH 8.0 with 1× Halt protease inhibitor cocktail (ThermoFisher Scientific)) for 15 min, homogenized with a pellet pestle (ThermoFisher Scientific), and spun at 450 x g for 5 min. Nuclei pellets were incubated in nuclear lysis buffer (1% SDS, 50 mM Tris-HCl pH 8.0, 10 mM EDTA with 1× Halt protease inhibitor cocktail) for 30 min, further lysed with a syringe, and sonicated with Covaris S2 to obtain fragments between 400 and 1000 base pairs. Fragment sizes were checked by running an aliquot of the sheared, purified chromatin on an agarose gel. Sheared chromatin was spun at 13,000 rpm for 10 min, and supernatant (2×10⁶ cells/reaction) was incubated overnight with 10 μg of ChIP-grade antibodies: anti-H3 (ab1791; Abcam); anti-HP1α (ab77256; Abcam); anti-H3K9me3 (ab8898; Abcam), anti-TRF2 (NB110-57130; Novus Biologicals); anti-TRF1 (ab1423; Abcam,) and anti-rabbit IgG (#2729; Cell Signaling). Samples were then immunoprecipitated with Dynabeads Protein G (Life Technologies) for >6 h to overnight, washed and eluted (1× TE, 1% SDS, 250 mM NaCl). Immunoprecipitated chromatin was treated with 0.2 μg/μl RNAse at 37 °C for 30 min, followed by reverse crosslinking (0.2 μg/μl Proteinase K (Bioline) and 200 mM NaCl) at 65 °C for >6 h to overnight. DNA was purified using NucleoSpin Gel and PCR cleanup kit (Macherey-Nagel), denatured (0.1 M NaOH) at 37 °C for 30 min, neutralized (6× SSC), and transferred to a Hybond-N+ membrane (Amersham) on a dot blot.

24 nt C-strand telomeric probes containing six ³²P-dC were synthesized[68]. One microliter annealed template C-rich oligo (1.7 pmol/μl), 1 μl of dTTP (1.25 mM

stock, final 50 μM), 7 μl $^{32}$P-dCTP (3000 Ci/mmol), 4 μl $^{32}$P-dATP (3000 Ci/mmol), 7.9 μl Millipore H$_2$O, and 1 μl Klenow (5 U/μl) were combined in a final volume of 25 μl. Room temperature extension was carried out for 30 min, and 95 °C for 5 min [to inactivate Klenow to prevent probe degradation upon uracil deglycosylase (UDG) treatment]. The reaction was cooled to room temperature. 0.5 μl UDG (1 U/μl) was added to degrade the GTU template, incubated at 37 °C for 15 min, and then UDG was inactivated at 95 °C for 10 min. Free isotopes were removed using an illustra microspin G-25 column (GE Healthcare, Piscataway, NJ). CENPB (5′-CTTCGTTGGAAACGGGA-3′) probes were end-labeled with [γ-$^{32}$P] ATP. Half of the blot was hybridized with C-strand telomeric probes, and the other half with CENPB probe at 42 °C overnight. The blots were then washed and exposed to a Phosphorimager screen (GE Healthcare). Uncropped dot blots are shown in Supplementary Fig. 1.

**Telomere restriction fragment length analysis.** Genomic DNA was purified using DNeasy Blood & Tissue kit (Qiagen, Valencia, CA). Telomere Restriction Fragment (TRF) length analysis was performed[69]. 1 μg purified genomic DNA was digested in 20 μl reaction with *Alu*, *Msp*I, *Hae*III, *Hin*fI, *Hha*I and *Rsa*I for 4 h at 37 °C. DNA was separated on a 0.8% agarose gel in 0.5× TBE. The gel was dried, denatured (0.5 M NaOH and 1.5 M NaCl for 1 h), rinsed with distilled water (3×), neutralized (0.5 M Tris-HCl pH8 and 1.5 M NaCl for 30 min), prehybridized (6× SSC, 5× Denhardt's solution, 0.5% (w/v) SDS), and hybridized with C-strand telomeric probe at 42 °C overnight. The gel was then washed and exposed to a Phosphorimager screen (GE Healthcare). Average overhang sizes were calculated using the formula mean average length = $\Sigma(\text{Int}_i) / \Sigma(\text{Int}_i / MW_i)$, where $\text{Int}_i$ = signal intensity and $MW_i$ = molecular weight of the DNA at position $i$[69].

DNA in Supplementary Fig. 6a and 6i were digested with *Rsa*I and *Hin*fI. Generally size markers were loaded on TRF gels. However, the DNAs in Supplementary Fig. 6a and 6i were only run for a very short time so that the telomeres would remain as compact as possible to maximize the ability to detect remaining telomere resulting from Cas9-digestion. As size markers would not have been able to be resolved during this short run, they were eliminated in this in-gel hybridization. Here we focused on quantifying telomeric intensity using Alu probe (5′-CACGGCCTGTAATCCCAGCACTTTG-3′) end-labeled with [γ-$^{32}$P] ATP as loading controls. Gels were denatured and neutralized between C-strand telomere probe and Alu probe hybridization. Uncropped gels are shown in Supplementary Fig. 10.

**Beta-galactosidase assay.** Senescence-associated beta-galactosidase (β-gal) was analyzed using colorimetric β-gal staining kit (Cell Signaling) or quantified by fluorometric kit (Cell Biolabs). Total protein was measured using Precision Red protein assay reagent (Cytoskeleton, Inc.).

**Cell growth assays.** Cells were infected with either WT or mutant hTRs at day 0, and selected with puromycin at day 2. Cells were split as needed to maintain logarithmic growth, and harvested at indicated time points and stained with trypan blue. Viable cells were scored by TC20 automatic cell counter (Bio-Rad).

**Telomere dysfunction-induced foci (TIF) image analysis.** Cells were washed with 1× PBS, fixed with 4% paraformaldehyde (w/v) (ThermoFisher) in 1× PBS and permeabilized with 0.5% NP-40 for 15 min. IF/FISH[42] was performed with modifications. For IF, cells were blocked (0.2% (w/v) fish gelatin, 0.5% (w/v) BSA in PBS for 20 min), and immunostained with the primary pAb anti-53BP1 (NB100-304; Novus Biologicals) 1:500 for 1 h. Cells were then washed and incubated with secondary antibody Alexa Fluor 488 (Molecular Probes) 1:750 for 1 h, fixed with 2% paraformaldehyde and incubated with 0.1 mg/ml RNAse for 1 h at 37 °C. For, FISH, cells were dehydrated sequentially with ethanol (70%, 95% and 100%; 5 min each), heated in hybridization mix with 0.5 mg/ml peptide nucleic acid (PNA) telomeric probe TelC-Cy3 (PNABio) at 85 °C for 10 min to denature the DNA, followed by overnight hybridization at room temperature. Nuclei were stained with DAPI (4,6-diamidino-2-phenylindole) (Life Technologies) and mounted with Prolong Gold (Invitrogen).

Equipment and settings: Images were captured using a DeltaVision Real-time Deconvolution Microscope (Applied Precision) with a ×100 oil 1.4 NA Plan Apo objective (Olympus) by a Photometrics CoolSNAP HQ monochrome CCD camera. 0.25 μm increments (×20 stacks for a total of 5 μm) were deconvoluted and Z-projected in SoftWoRx (Applied Precision).

TIFs colocalization analysis: Z-Projected images were converted to Tagged Image File Format (TIFF) using the Fiji image processing package (www.fiji.sc). Enumeration of 53BP1 and telomeric foci were quantified using CellProfiler 2.1.1. (www.cellprofiler.org) image analysis software. For foci scoring, identical thresholds were applied to all controls and experimental groups, followed by colocalization (TIFs) masking (pipelines available on request).

**TRF2 knockdown.** Cells were transfected with ON-target plus smart pool consisting TRF2 (siRNA) or si-non-targeting (Dharmacon) using Lipofectamine RNAiMAX reagent (Life Technologies) following manufacture protocols, and analyzed at ~72 h.

**STORM image acquisition and analysis.** STORM equipment and settings: STORM[70] was performed on a custom-built microscope based on a Nikon Ti-U inverted microscope. Three activation imaging lasers (Coherent CUBE 405, OBIS 561 and CUBE 642) were combined using dichroic mirrors, aligned, expanded and focused to the back focal plane of the objective (Nikon Plan Apo ×100 oil NA 1.45). The lasers were controlled directly by the computer. A quad band dichroic mirror (zt405/488/561/640rpc, Chroma) and a band-pass filter (ET705/70m, Chroma) separated the fluorescence emission from the excitation light. During image acquisition, the focusing of the sample was stabilized by a closed-loop system that monitored the back reflection from the sample cover-glass via an infrared laser beam sent through the edge of the microscope objective.

A low-end piezoelectric deformable mirror (DM) (DMP40-P01, Thorlabs) was added in the emission path at the conjugate plane of the objective pupil plane[70]. By first flattening the mirror and then manually adjusting key Zernike polynomials, this DM corrected aberrations induced by both the optical system and the glass-water refractive index mismatch when the sample was several micrometers away from the coverglass. After correcting these aberrations, an astigmatic aberration was further added by the DM for 3D STORM. The fluorescence was recorded at a frame rate of 57 Hz on an electron multiplying CCD camera (Ixon+ DU897E-CS0-BV, Andor).

The mounting medium used for STORM imaging was PBS with the addition of 100 mM mercaptoethylamine at pH 8.5, 5% glucose (w/v) and oxygen scavenging enzymes 0.5 mg/ml glucose oxidase (Sigma-Aldrich), and 40 mg/ml catalase (Roche Applied Science). The buffer remained suitable for imaging for 1–2 h. Photo-switchable dye Cy5 was used for imaging with a ratio of one dye per PNA probe. Cy5 was excited with a 642 nm imaging laser, with a typical power at the back port of the microscope being 30 mW. Analysis of STORM raw data was performed in the Insight3 software[47], which identified and fitted single molecule spots in each camera frame to determine their x, y, and z coordinates as well as photon numbers. Sample drift during data acquisition was corrected using imaging correlation analysis. The drift-corrected coordinates, photon number, and the frame of appearance of each identified molecule were saved in a molecule list for further analysis.

STORM imaging: Cells were labeled with PNA telomeric probe, TelC-Cy5 (PNABio). Individual telomeric localization signals were detected by switching the fluorophores between active and dark states stochastically. Accumulation of individual fluorophore forms a cluster of molecular positions, known as localizations, corresponding to structural characteristics of an individual telomere.

STORM analysis: Individual telomeres were manually selected from the STORM images. The telomeres near focal planes with good resolution were picked. These manually picked telomeres were further screened so that telomeres with more than 200 localizations were kept for the Radius of gyration (Rg) analysis.

**Statistical analyses.** Significance of mean was assessed by statistical analyses noted in the corresponding figure legends. These include: one-way ANOVA and Dunnett's multiple comparison test with 95% confidence level; two-tailed unpaired $t$ test with 95% confidence level. All graph bars were represented by means with standard error of the mean (s.e.m.). For STORM statistical analysis, means of Rg in the violin plots were compared using ANOVA Tukey's multiple comparisons with 95% confidence level.

**Code availability.** Custom image analysis for Rg calculation were written in MATLAB 2012B. The MATLAB script is available from the authors upon request.

## Data availability

All relevant data and supplementary information files are included in this published article. All other supporting information is available from the authors upon reasonable request.

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

## Acknowledgements

This work was supported in part by the National Institutes of Health National Cancer Institute grant CA096840 (E.H.B.), UCSF Brain Tumor SPORE Developmental Project Grant (E.H.B.), W.M. Keck Foundation Medical Research Grant (B.H.), and the Damon Runyon Cancer Research Foundation Postdoctoral Fellowship DRG2168-13 (T.T.C.). B.H. is a Chan Zuckerberg Biohub investigator. We thank Geeta Narlikar, John Murnane, Jue Lin, and Dana Smith for the stimulating scientific discussions and critical edits of the manuscript; Barbara Panning for insightful scientific discussions; Bradley Stohr, Lifeng Xu, Tom Misteli, and Adam Larson for reagent sharing; Beth Cimini, Stephanie Johnson, and Yina Wang for discussions on image analyses; Oria Lu for technical help.

## Author contributions

T.T.C. and E.H.B. designed the experiments. T.T.C., X.S. and E.H.B. wrote the manuscript; T.T.C., J.-H.W. and G.S. performed the experiments; X.S. and J.G. performed the STORM imaging. T.T.C., X.S, J.G., G.S., B.H. and E.H.B. analyzed the data. All authors provided feedback on the manuscript.

## Additional information

**Competing interests:** The authors declare no competing interests.

