## [Peer Review File · Nature Communications]

Reviewers' comments:

Reviewer #1 (Remarks to the Author):

The manuscript entitled "Local Enrichment of HP1 α at Telomeres Alters Structure and Telomere Extension" by Tracy Chow and colleagues reports the effects of artificially imposing heterochromatin on telomere biology. For that, the authors create a TRF1-HP1 fusion protein aimed at specifically tethering HP1 at telomeres. They show that constitutive tethering of HP1 to TRF1 not only enriches HP1 at telomeres but also results in the increase of local H3K9me3, even though it results in the overall reduction of telomeric histone H3. In addition, TRF1-HP1 reduces the activity of the telomerase, an effect that is mediated by the Cromo Shadow Domain. Finally, using STORM technology the authors shown that the TRF1-HP1 fusion induces alteration of the 3D structure of chromosome-ends with the formation of irregular shapes.

This manuscript is well structured and clear. It possesses the merit of using a chimeric protein (with all its potential artefacts) intended to induce heterochromatin specifically at telomeres. The use of this novel method along with high quality data will certainly be of interest to the telomere community. Nevertheless, some issues should be addressed before publication.

Major Points:

1. The major unexplained issue in the manuscript is the failure to explain how over-expressed TRF1-HP1, despite capable of inhibiting telomerase function *in vivo*, fails to modify telomere length over several population doublings (Supp. Figure 2b). Is there a compensation of HP1 heterochromatin at telomeres on longer PDs in these cells? Is the lack of effects observed in primary cells (Fig. 2f-h) a reflection of this potential compensation?
2. Even though the authors claim that "TRF1HP1 α is expressed and specifically enriched at telomeres." they fail to evaluate the potential effects of tethering TRF1-HP1 to the remaining HP1 genomic loci. Could it be possible that the presence of TRF1 at these sites has an additional effect that could also interfere with telomerase activity.
3. In Fig 3d, It appears that the mutation in HP1 CD (V22M) causes a reduction of TRF1-HP1 V22M foci co-localizing with TRF2 and so a reduction of the TRF1-HP1 V22M at telomeres. Is this statistically significant? Is there a dependency on TRF1-HP1 binding to H3K9me3 at telomeres? Or a reduced number of TRF2 foci without TRF1-HP1 co-localization?
4. In Fig.4b, is it possible that the increase of the TIFs in the line TRF1-HP1 mutated in the CD (V22M) compared to TRF1-HP1 WT is related with the observed decreased co-localization with TRF2 (fig.3d).
5. As a potential mechanistic insight, the observed increase of H3K9me3 heterochromatin at telomeres may result in reduced expression of the telomere non-coding RNA TERRA, perhaps interfering with telomerase recruitment. Are TERRA levels altered upon TRF1-HP1 overexpression?

Minor points:

In Fig.1f, the quantified differences that emerges from the dot blots shown are slight. It would be worth to present as a supplemental figure the remaining replicates. It would also be important to add information on potential changes of TRF1 levels at telomeres and centromeres upon expression the TRF1-HP1 fusion protein.

In Fig. 1j, TRF2 levels at telomeres normalized to H3 upon expression of TRF1-HP1 show an unexpectedly high SEM compared with the remaining experiments. It would be important to show individual experiments to evaluate the potential source of variability.

Reviewer #2 (Remarks to the Author):

This study addresses the question of the role of heterochromatin in telomere function. It describes a new system of HP1 α targeting through the telomere binding protein TRF1. Upon overexpression

and incorporation of TRF1-HP1a proteins, the ability of telomere to elongate when the telomerase RNA component hTR is overexpressed is impaired. A separation of function analysis of HP1 reveals that this effect is due to functions of the CSD that appear independent of the dimerization/oligomerization capacity of HP1 and that this telomerase elongation negative regulation can be uncoupled from a telomere protective function.

Overall, the quality of the data is excellent and convincing. However, this work greatly suffers from the artificial nature of the targeting/overexpressing system that prevents any firm conclusion on the biological significance of the results. Thus, as such, this work appear potentially interesting but preliminary. Additional experiments are mandatory to really interpret the results in term of general telomere biology. Below, find some suggestions.

- Is the TRF1-EGFP-HP1 fusion functional in term of TRF1 function. For instance, could it rescued the MTS formed upon TRF1 downregulation ?
- What is really the level of overexpression and telomere incorporation of TRF1-HP1 as compared to non overexpressed control cells: this implies the use of TRF1 antibodies in WB and chip instead of GFP antibodies.
- The phenotype of telomere elongation and protection must be studied in a setting where TRF1-HP1 is expressed at a nearly physiological level (in combination with RNA interference strategy)
- In Figure 4, the interpretation of the separation of function mutants of HP1 regarding telomere elongation inhibition and increased protection in the context of mutant hTR expression is rather uncertain since it based on the assumption that the hTRT mutant repeat will be behave similarly regarding HP1 inhibition. In order to clearly show the telomere protective effect of HP1, other telomere dysfunction setting (such as TRF2 or POT1 downregulation) should be studied.
- What is the level of HP1 naturally present at telomeres in the cell line studied? How much it is increased upon TRF1-HP1? What happens if the expression of the endogenous HP1 gene is downregulated?
- The effect of the HP1 mutants impairing telomere elongation should be studied in a cell line engineered to express the mutants from the endogenous HP1 locus.

Minor

- The increment in H3K9me3 upon TRF1HP1 overexpression is very little and not really visible from the slotblot of figure 1f.

Reviewer #3 (Remarks to the Author):

In Chow et al. the authors describe a novel use of TRF1 as tethering tool to enrich specific proteins at telomeres, in this case HP1a WT and mutant variants. Through this approach, the authors show that HP1a enrichment at telomeres can counteract telomere lengthening in hTR over-expression conditions and can induce alterations in telomere structure. The findings described are interesting for the field of telomeres and HP1 function. In particular, the use of HP1a point mutants allows to uncouple different activities of HP1a, highlighting that H3K9me3 binding is dispensable for the observed phenotypes and suggesting that the dimerization of HP1a and the binding of an unknown ligand are involved. Although these observations are exciting the work scratches the surface of the molecular events behind them. Further characterisation of the HP1a ligands that prevent telomere lengthening and mediate telomere reorganisation would greatly reinforce the message of this work. Beside this aspect that could be beyond the scope of this work, additional points should be addressed to fully support the conclusions proposed.

Major points:

- Most importantly, the authors don't discern in a clear way the effects related to TRF1-HP1a tethering to telomeres to the ones related to TRF1 overexpression itself. Indeed, the overexpression of TRF1 alone causes a slight decrease of H3 compared to Vonly (ChIP, Fig 1h) and an increase in Rg and average number of localisations per telomere compared to HP1a (STORM,

Fig 5h, line 260). Also, "long term culturing conditions of TRF1 overexpression in certain cancers resulted in telomere shortening³⁹" (line 161). Given this, Vonly controls should be included in STORM analyses and every experimental group (Vonly, TRF1 and HP1a) should be considered separately to compute statistics in ChIP experiments.

- Although TRF1-HP1a tethers HP1a to telomeres, TRF1 will in turn be tethered to HP1a binding sites. The use of mutants only partially addresses this point. The authors should discuss the potential implications of the aberrant delivery of TRF1 to non telomeric regions. Do the authors envisage technological improvements to overcome this limitation?

- Colocalisation analyses: how does colocalisation is measured in this study? A methodological description should be added in the Methods section. Also, the extent of colocalisation between TRF2 and HP1 is very high. How the authors explain this observation? A short sentence is included in the figure legend but it is hard to verify in the images. To this regard, bigger images or zoom in panels would help the interpretation of results.

- ChIP experiments: in the introduction, the authors mention that knockout of SUV39H1/2 and SUV4-20H1/2 result in defective telomere function and telomere lengthening. It would be interesting to verify by ChIP if H4K20me3 levels and/or SUV4-20H recruitment is affected by TRF1-HP1a. Indeed, direct binding between HP1a has been described (Schotta G et al Genes & Dev 2004; Hahn M et al Genes & Dev 2013) and the binding contributes to chromatin compaction at centromeres through recruitment of Cohesin (Hahn M et al Genes & Dev 2013). Also, SUV4-20H deficiency results in telomere lengthening in MEFs and mESCs (Benetti R et al JCB 2007). The involvement of SUV4-20H in the phenotype observed would be an intriguing possibility and observing an increase in H4K20me3 would reinforce the concept of heterochromatin mark accumulation shown by a slight increase in H3K9me3.

- STORM experiments: the authors calculate the Rg of telomeres to characterise alterations in telomere shape. This type of analysis requires very accurate drift correction as suboptimal drift correction can lead to deformation of telomeres in super-resolution. Also, the low number localisation/frame typically found in this type of imaging, can make drift correction challenging. Do the authors use beads? If this is the case it should be mentioned. If beads were not used, authors should confirm their observations in the presence of beads. Also, how telomeres are identified and analysed? Are they manually selected? Are they automatically identified with clustering algorithms? Please provide this information in the methods section.

Since TRF1 and HP1a controls show different Rg values and average number of localisations, a Vonly control should be introduced to determine the Rg in a more neutral control condition and to define whether TRF1 overexpression alters telomere structure.

Minor points:

- The authors should clearly cite the references referring to each of the mutants described in the manuscript and include in the methods section the source of the constructs or how they were generated.

- Telomere length analyses in Figure 2b and especially in Supplementary Figure 2b are not clearly visible. Higher exposure images should be shown.

- Please add molecular weight to Supplementary Figure 1a.

- Adding colours to Supplementary Figure 3b would help the visual interpretation of the graph

- Line 257: "we only analysed telomere clusters with centers of mass near the focal plane, and consisting of more than 200 localization points (Fig. 5c)." Figure 5c does not refer to this.

- Line 259: "The average number of localization points of such filtered individual telomeres for TRF1, HP1a, WT TRF1HP1a, TRF1HP1aI165A were 664, 420, 542, and 640, respectively." Please add standard deviation values to include information about the variability of localisations detected.

- The authors should include a section describing the statistical methods applied in the methods section.

- TRF1-HP1a and senescence: the authors show that TRF1-HP1a does not exacerbate the levels of B-Gal positive cells in high passage BJ and WI38 cells. Could TRF1-HP1a instead anticipate the appearance of B-Gal positive cells when overexpressed in low passage cells?

- Lines 549 and 550: the sentence does not refer to STORM imaging, please check and move to correct section.
- In the Discussion the main biological impact of their findings should be stressed out. Also the discussion about TRF2 might be reduced or eliminated, considering the results presented.

Responses to the Review Comments

Manuscript #: NCOMMS-18-00098A

Revised Title: Local Enrichment of HP1alpha at Telomeres Alters Their Structure, Protection and Extension by Telomerase

Authors: Tracy T. Chow, Xiaoyu Shi, Jen-Hsuan Wei, Juan Guan, Guido Stadler, Bo Huang, Elizabeth H. Blackburn

Date: May 16, 2018

We are appreciative of the very thoughtful and constructive reviews we received on the original manuscript. We were heartened to read that Reviewer 1 found this work “possesses the merit of using a chimeric protein intended to induce heterochromatin specifically at telomeres,” and “well structured and clear,” and “The use of this novel method along with high quality data will certainly be of interest to the telomere community”, that Reviewer 2 indicated this work described “a new system of HP1a targeting through the telomere binding protein TRF1”, and that “the quality of the data is excellent and convincing,” and that Reviewer 3 found the data to be “a novel use of TRF1 as tethering tool to enrich specific proteins at telomeres,” and “The findings described are interesting for the field of telomeres and HP1 function.”

Below, please find our responses to all of the reviewers’ other comments. We have added substantial new data to help allay their key concerns and thereby also significantly strengthen our main conclusions. We hope that our clarifications and the new data help make the revised manuscript more coherent and complete. Our point-by-point responses to reviewers’ comments are delineated in blue.

Reviewer #1 (Remarks to the Author):

The manuscript entitled “Local Enrichment of HP1alpha at Telomeres Alters Structure and Telomere Extension” by Tracy Chow and colleagues reports the effects of artificially imposing heterochromatin on telomere biology. For that, the authors create a TRF1-HP1 fusion protein aimed at specifically tethering HP1 at telomeres. They show that constitutive tethering of HP1 to TRF1 not only enriches HP1 at telomeres but also results in the increase of local H3K9me3, even though it results in the overall reduction of telomeric histone H3. In addition, TRF1-HP1 reduces the activity of the telomerase, an effect that is mediated by the Chromo Shadow Domain. Finally, using STORM technology the authors shown that the TRF1-HP1 fusion induces alteration of the 3D structure of chromosome-ends with the formation of irregular shapes.

This manuscript is well structured and clear. It possesses the merit of using a chimeric protein (with all its potential artefacts) intended to induce heterochromatin specifically at telomeres. The use of this novel method along with high quality data will certainly be of interest to the telomere community. Nevertheless, some issues should be addressed before publication.

RESPONSE: We are glad the reviewer thinks the method is novel and the work will be of interest to the telomere community. We also recognize the reviewer’s concern about the potential artifacts of the chimeric system, and address these and other concerns below.

Major Points:

1. The major unexplained issue in the manuscript is the failure to explain how over-expressed TRF1-HP1, despite capable of inhibiting telomerase function in vivo, fails to modify telomere

length over several population doublings (Supp. Figure 2b). Is there a compensation of HP1 heterochromatin at telomeres on longer PDs in these cells?

RESPONSE: We thank the reviewer for drawing our attention to this point, which we can clarify, as follows. Telomere elongation was inhibited where there was non-limiting hTR (i.e. hTR overexpression) in UM-UC3 (Fig 2b-c). We were able to take advantage of the fact that in the first few days after we introduced higher non-limiting hTR, the telomere length was dynamic (i.e., growing). We used this telomeric-length-growing phase to experimentally scrutinize telomere length regulation. We do concur with the reviewer, that in situation of the absence of additional hTR, because UM-UC 3 telomeres are quite short, it is certainly possible to have compensation of HP1 α heterochromatin at telomeres across multiple population doublings, and / or selective pressure against cells with shortened telomeres. We have now clarified the interpretations within the legend of old Supplementary Fig. 2b (now renumbered as Supplementary Fig. 3b).

On the other hand, the hTR overexpression in UM-UC3 cells alleviates selective pressure on cells with critically short telomeres. The hTR overexpression also serves to increase the signal to noise ratio regarding telomerase action. Thus, this further supports the idea that the effect of TRF1HP1 α is mediated through effects on telomerase.

Is the lack of effects observed in primary cells (Fig. 2f-h) a reflection of this potential compensation?

RESPONSE: The senescence marker (Beta gal staining) experiments in primary cells were performed in a much shorter time frame (assayed on day 10-12 days after infection). In addition, the mean population doubling (PD) time of aged fibroblasts can be up to ~48hrs (compared to ~24hrs in UM-UC3 cells). Thus, it is less likely that the lack of beta-gal change was due to compensation. This is because during this short time, only a gradual and small shortening of telomere length is expected. The dropping out of a certain small fraction of cells will not have much effect on the apparent PD rate. However, we agree with the reviewer that one cannot exclude this possibility. We have now clarified the interpretations in the legend of Fig. 2f.

2. Even though the authors claim that “TRF1HP1 α is expressed and specifically enriched at telomeres.” they fail to evaluate the potential effects of tethering TRF1-HP1 to the remaining HP1 genomic loci. Could it be possible that the presence of TRF1 at these sites has an additional effect that could also interfere with telomerase activity.

RESPONSE: We agree with the reviewer that it is important to rule out indirect effects of TRF1HP1 α localization at sites other than telomeres, and thank the reviewer for prompting us in the revised text to better highlight our results which do address this point. In our previous submission, we particularly made sure to carry out several control experiments (described below) to rule out the possibility of indirect effects.

We respectfully point out that “TRF1HP1 α is expressed and specifically enriched at telomeres.” is not merely a claim, but rather a strong set of evidence. We provided multiple imaging data (Fig. 1b, d; Fig. 3b, d) showing indeed TRF1HP1 α is expressed and specifically enriched at telomeres, but not exclusively except for the V22M mutant and double mutant V22MI165A.

Concerning the reviewer's point about "fail to evaluate the potential effects tethering TRF1-HP1 to the remaining HP1 genomic loci", our mutational approach allowed us to specifically address this very good point. **1)** We deliberately used V22M to control for possible indirect effects of TRF1 being recruited to other genomic sites by HP1 α . While WT-TRF1HP1 α was observed, as expected, to be localized to other genomic foci via H3K9me3 binding to such sites (Fig. 3b, c), mutant V22M (which fails to recognize H3K9me3, a major recruitment mechanism to other genomic sites), as expected, lost its ability to bind to heterochromatin marks at non-telomeric genomic regions. We accordingly observed that mutant V22M was exclusively tethered by TRF1 fusion at the telomeres, and not to other regions in the genome (Fig. 3b-d). **2)** WT-TRF1HP1 α and mutant V22M limited telomerase activity to similar extents (Fig 3f, h). We also included TRF1 alone or HP1 α alone as additional negative controls (Fig 2b-c). **3)** We have now also included additional data showing there was no significant difference of "# TRF2 per nucleus" between WT-TRF1HP1 α and its corresponding mutants (new panel, Fig. 3e), indicative of preserved telomere integrity.

Hence, it was not the association of TRF1HP1 α to other genomic sites that caused the effect on telomerase action in cells. Thus, this phenotype is unlikely to be confounded by indirect effects. We realize that our description of these experiments was not clear, and have more clearly emphasized the rationale and results of these control experiments within the revised main text (lines 173-176; 208-213).

3. In Fig 3d, it appears that the mutation in HP1 CD (V22M) causes a reduction of TRF1-HP1 V22M foci co-localizing with TRF2 and so a reduction of the TRF1-HP1 V22M at telomeres. Is this statistically significant?

RESPONSE: We thank the reviewer for these thoughtful comments, and we can clarify our interpretation of the colocalization of EGFP-TRF1HP1 α mutants with TRF2 in these images. Because V22M and V22MI165A mutants have lost the ability to recognize H3K9me3, as predicted these TRF1 fusion proteins did not bind to other genomic regions (Fig. 3b), and hence showed the expected reduced total % area of fusion protein per nucleus (Fig 3c) and fewer total fusion protein spots per nucleus (Supplementary Fig. 4). Thus, for the V22M or V22MI165A mutants, in these images (which are 2D projection images of z-stack images), any apparent reduction of their EGFP-fusion protein colocalization with TRF2 is likely to be (at least) partially driven by less random overlap of TRF2 telomere spots with HP1 α fusion protein bound to other genomic regions. Comparing to WT-TRF1HP1 α as the baseline, the difference between WT-TRF1HP1 α and V22M reached statistical significance (**p=0.008), although double mutant V22MI165A showed no statistically significant difference. Using V22M as baseline there was no significant difference between V22M vs V22MI165A (one-way ANOVA Dunnett's multiple comparison test; 95% confidence level).

We have now added new experimental data in Fig 3e and Supplementary Fig. 4. First, there is no significant difference of total number of TRF2 foci among all the groups (Fig. 3e). Secondly, Supplementary Figure 4 shows that, as predicted for random overlap of telomere spots with other bound genomic loci in these images, indeed we detected a significantly higher frequency TRF1HP1-WT spots compared to -V22M or -V22MI165A spots. We have updated the figure legends of Fig 3d to clarify this point. Thus, here we focus on the fact that TRF1HP α and its various mutants were all capable of efficiently localizing at the telomeres.

Is there a dependency on TRF1-HP1 binding to H3K9me3 at telomeres? Or a reduced number of TRF2 foci without TRF1-HP1 co-localization?

RESPONSE:

“Is there a dependency on TRF1-HP1 binding to H3K9me3 at telomeres?”

In our experimental set up, the binding of TRF1HP1 α to telomeres will be potentially mediated by HP1 α recruitment to H3K9me3¹ as well as dependent on fusion to TRF1. Our data showed a chromo domain mutation (V22M) within HP1 α in TRF1HP1 α , even though causing the fusion protein to be unable to recognize H3K9me3, was as efficiently tethered at the telomere when fused with TRF1 as WT-TRF1HP1 α (Fig 3b, d). Thus, the loss of any recruitment of HP1 α to telomeres that might have occurred via H3K9me3 recognition was compensated for by the fusion to TRF1.

“Or a reduced number of TRF2 foci without TRF1-HP1 co-localization?”

Given TRF1 efficiently goes to telomeres, the reduced number of colocalization spots of V22M with TRF2 can be accounted for by the lack of V22M being targeted to other genomic regions, and consequently the observed reduction in random overlap spots with TRF2. We have added new supplemental data (Fig. 3e) indicating no significant difference in the number of total TRF2 foci per nucleus among TRF1HP1 α or its mutants, including -V22M, -I165A or double mutant -V22MI165A.

Re-iterating in this context what was described above, WT-TRF1HP1 α and all other mutants tested here have considerable amounts of localizing to other genomic regions compared to V22M or V22MI165A (which cannot recognize H3K9me3). Thus, compared to WT-TRF1HP1 α , V22M and V22MI165A showed significantly reduced % area of fusion protein per nucleus (Fig 3b, c) and reduced total numbers of fusion protein spots per nucleus (Supplementary Fig. 4) because of they are unable to bind other genomic regions (Fig. 3b). Thus, the resulting fewer random overlaps of such other genomically located HP1 α spots with telomere spots can account for the observation of fewer TRF2 spots co-localizing with TRF1HP1 α -V22M or -V22MI165A spots.

4. In Fig.4b, is it possible that the increase of the TIFs in the line TRF1-HP1 mutated in the CD (V22M) compared to TRF1-HP1 WT is related with the observed decreased co-localization with TRF2 (fig.3d).

RESPONSE: We thank the reviewer for this good point. However, the difference of %TIFs between WT TRF1HP1 α versus its mutant V22M in Fig. 4b were not statistically significant, so we believe any speculation about the slight experimental variations between these groups would be premature. Regarding the apparent decreased colocalization with TRF2 (Fig 3d), please refer to our response above to reviewer’s comment #3, and the updated figure legend of Fig 3d.”

5. As a potential mechanistic insight, the observed increase of H3K9me3 heterochromatin at telomeres may result in reduced expression of the telomere non-coding RNA TERRA, perhaps interfering with telomerase recruitment. Are TERRA levels altered upon TRF1-HP1 overexpression?

RESPONSE: We thank the viewer for this very interesting suggestion. However, despite extensive previous work, interpretations of the biological function(s) of TERRA have

unfortunately remained challenging². In addition, reports on TERRA analyses have also been contradictory. Some reported telomere length is regulated by TERRA³ while others reported no correlation between telomere length and TERRA levels^{4, 5}. Complications in TERRA analyses are likely due to technical draw backs in various technical methodologies^{2, 6}. Hence it would be premature to include observations on any alteration of TERRA level that might be detected into our working model. Any interpretation could be at best speculative given the lack of precise TERRA quantification methods. We agree that studying TERRA is interesting; however, this is tangential to the key focus of our main conclusions, and would require a new project beyond the scope of focus of this manuscript.

All of the minor text issues mentioned below have been incorporated in the revised manuscript.
Minor points:

In Fig. 1f, the quantified differences that emerges from the dot blots shown are slight. It would be worth to present as a supplemental figure the remaining replicates.

Replicates of all whole, uncropped dot blots are now included (Supplementary Fig. 1).

It would also be important to add information on potential changes of TRF1 levels at telomeres and centromeres upon expression the TRF1-HP1 fusion protein.

We have now included ChIP data using TRF1 antibody to detect the potential changes of TRF1 levels at telomeres using centromeres as controls (Supplementary Fig. 1d). TRF1 expression alone showed an average of ~2.5 fold increased TRF1 at telomeres compared to Vonly or HP1 α alone controls. We suspect the decreased TRF1 signal at the telomeres in TRF1HP1 α is likely due to inhibition of antibody accessibility caused by particular protein configurations of TRF1HP1 α (Supplementary Fig. 1d). Therefore, given limited options of available ChIP-grade antibodies, we could not evaluate the TRF1 signal of TRF1HP1 α .

In Fig. 1j, TRF2 levels at telomeres normalized to H3 upon expression of TRF1-HP1 show an unexpectedly high SEM compared with the remaining experiments. It would be important to show individual experiments to evaluate the potential source of variability.

We have added all replicates of raw ChIP dot blots (Supplementary Fig 1). The high standard error of the mean (SEM) range was due to one replicate having a relatively higher detected TRF2 signal. Please note that these TRF2 data do not change the working model of the manuscript.

Reviewer #2 (Remarks to the Author):

This study addresses the question of the role of heterochromatin in telomere function. It describes a new system of HP1a targeting through the telomere binding protein TRF1. Upon overexpression and incorporation of TRF1-HP1a proteins, the ability of telomere to elongate when the telomerase RNA component hTR is overexpressed is impaired. A separation of function analysis of HP1 reveals that this effect is due to functions of the CSD that appear independent of the dimerization/oligomerization capacity of HP1 and that this telomerase elongation negative regulation can be uncoupled from a telomere protective function.

Overall, the quality of the data is excellent and convincing. However, this work greatly suffers

from the artificial nature of the targeting/overexpressing system that prevents any firm conclusion on the biological significance of the results. Thus, as such, this work appears potentially interesting but preliminary. Additional experiments are mandatory to really interpret the results in term of general telomere biology. Below, find some suggestions.

RESPONSE: We are happy the reviewer finds the data excellent and convincing. While we recognize the reviewer's concerns about the artificial nature of our targeting system, we feel that, in particular our sets of control experiments comparing well-characterized HP1 mutants do help isolate the biological significance of our results, as described below.

We would note that our system had the following strengths:

- A. The locus-tethering approach (and the mutants) allowed us to manipulate chromatin dynamics specifically at the telomeres, not confounded by challenges to interpret molecular mechanisms of the dynamics of telomeric chromatin and telomerase action in settings where genome-wide chromatin alterations also take place.
- B. In addition, we observed and quantified multiple parameters of individual telomeres by use of various microscopic techniques (conventional fluorescence and STORM).
- C. These approaches revealed HP1 α allele-specific functions, including a relationship between the control of telomerase action, protection and telomeric structural changes.

In this context, we thank the reviewer for the additional suggestions to strengthen our conclusions, and have added new experiments to address these suggestions below.

- Is the TRF1-EGFP-HP1 fusion functional in term of TRF1 function. For instance, could it rescued the MTS formed upon TRF1 downregulation?

RESPONSE: We thank the reviewer for this interesting experimental suggestion, and the opportunity to address this question via below two alternative experimental approaches; both indicate that TRF1 remains functional in the context of TRF1HP α .

1) For reviewer only: The quantification of multi-telomeric signals (MTS)^{7, 8} showed TRF1HP α resulted in slightly less MTS compared to Vonly. This difference was seen under four different independent experimental settings, namely, when expressing:

- a) empty vector,
- b) WT hTR,
- c) mutant hTR 47A
- d) mutant hTR TSQ1

~8-16 metaphase spreads were manually analyzed in each group. Thus, for statistical analysis we combined the results of all 4 experiments together. There is a slight but significant difference: Compared to Vonly, TRF1HP α was consistently associated with a lower frequency of multi-telomeric signals per chromatid end in metaphases. *p=0.0246 (unpaired student t-test with 95% confidence interval; error bars represent SEM). This implies tethered TRF1HP α does not cause, and may even confer protection from the telomeric DNA replication-induced damage that is thought to cause the appearance of multi-telomeric signals at a chromatid end^{7, 8}.

2) CRISPR experiment (Supplementary Fig 6i-j): Expressing TRFHP1 α was as efficient as expressing TRF1 alone in protecting telomeres from telomeric DNA cutting by telomeric DNA-targeted CRISPR/Cas9, performed in intact UM-UC3 cells.

- What is really the level of overexpression and telomere incorporation of TRF1-HP1 as compared to non overexpressed control cells: this implies the use of TRF1 antibodies in WB and chip instead of GFP antibodies.

RESPONSE: We thank the reviewer for bringing up a good point that we can address:

For ChIP, we have now included results for TRF1 antibody (Supplementary Fig. 1a, b, d) showing an average of ~2.5 fold increased TRF1 at telomeres in TRF1 compared to Vonly or HP1 α . However, TRF1HP1 α shows weaker TRF1 antibody signal at the telomeres. As expected TRF1 antibody immunoprecipitation yielded a much stronger signal with telomere probe compared to centromere probe. Moreover, HP1 α is indeed efficiently tethered at the telomeres by TRF1HP1 α (Fig 1f-g; Supplementary Fig 1); thus, we suspect the weaker TRF1 antibody signal at the telomeres in TRF1HP1 α is likely caused by limited accessibility of TRF1 antibody due to the particular configurations of TRF1HP1 α . Given limited options of available ChIP-grade antibodies, we could not evaluate the TRF1 signal of TRF1HP1 α .

For western analyses, we tested alternative antibodies using TRF1 cDNA overexpression. Of the several antibodies we tested, only the ChIP-grade antibody can detect both reported TRF1 variants (NM_017489 and NM_003218).

For reviewer only: Here we show an example of a TRF1 antibody (left) not able to detect TRF1 variant 2. In total, four different commercially available TRF1 antibodies were tested, and only the ChIP-grade antibody detected variant 2, which is associated with our TRF1HP1 α . Thus, we reverted back to ChIP-grade antibody (right) for western analyses (Supplementary Fig. 2b, e). Overexpression of TRF1 and TRF1HP1 α mutants were readily detected with undetectable endogenous TRF1. Though western analyses showed a considerable amount of overexpression, only ~2.5 fold increase of TRF1 are localized at the telomeres (Supplementary Fig. 1a, b, d).

- The phenotype of telomere elongation and protection must be studied in a setting where TRF1-HP1 is expressed at a nearly physiological level (in combination with RNA interference strategy).

RESPONSE: We thank the reviewer for the comments, and the opportunity to clarify our experimental design strategies. While some HP1 α has been found naturally occurring at human telomeres, it is at rather low frequency⁹⁻¹³. Our rationale for using TRF1HP1 α therefore was to increase the probability of targeting HP1 α specifically to telomeres for reliable quantitative analyses. While the tethering approach allows us to achieve this goal, this makes it difficult to make a direct comparison with physiological levels of HP1 and TRF1 because the fusion protein construct effectively increases the local concentration of HP1 at telomeres. We do agree with the reviewer that we recognize that the expression of this chimeric construct may have non-

specific or indirect effects. To control for these effects, we carefully carried out multiple sets of control experiments, as follows:

- 1) TRF1HP1 α overexpression did not cause increased baseline TIFs (Fig. 4f)
- 2) TRF1HP1 α does not significantly alter TRF2 occupancy at the telomeres (Fig 1f, j). Thus, TRF1HP1 α overexpression is not causing some significant disruption of telomere integrity.
- 3) There was only minimal change in cell growth rates (Supplemental Fig. 5)
- 4) We included multiple negative controls: Vonly, TRF1 alone and HP1 α alone are all in the same vector expression system (Fig. 2a-c). Moreover, we also have additional controls for TRF1HP1 α by making point mutations within HP1 α , which we showed were expressed at generally similar levels across experimental groups (Supplementary Fig. 2). Comparing point mutations one by one with each other, we observed clear, allele-specific differences in telomere extension (Fig 3f-i). Specific point mutations in TRF1HP1 α abrogated (completely in some cases) TRF1HP1 α -WT's effect on telomerase action. Thus, overexpression per se cannot cause the observed differences in telomerase regulation in WT TRF1HP1 α versus the point mutants of TRF1HP1 α .

We have updated the rationale description in the main text (lines 91-98) to clarify that the goal of our tethering approach was to strengthen the presence of this naturally occurring component (i.e. HP1 α) of telomeric chromatin in order to investigate and hence understand its role in telomere biology.

- In Figure 4, the interpretation of the separation of function mutants of HP1 regarding telomere elongation inhibition and increased protection in the context of mutant hTR expression is rather uncertain since it based on the assumption that the hTR mutant repeat will behave similarly regarding HP1 inhibition.

RESPONSE: Regarding reviewer's comments on "hTR mutant repeat will behave similarly regarding HP1 inhibition", as we understand it, this reviewer's phrase is referring to the potential for a different incorporation efficiency via telomerase enzymatic action (of mutant hTR compared to WT hTR upon tethering HP1 α at telomeres), and we thank the reviewer for the comment that we can clarify.

To assess how well the HP1 α mutants could protect from telomere damage, our first approach was to co-express mutant hTRs with separation-of-function HP1 α fusion protein mutants. We tested two independent mutant hTRs harboring different sequences (47A or TSQ), both known to induce DNA damage at telomeres via the incorporation of mutant DNA repeats that cannot bind telomeric DNA sequence-specific shelterin components normally. While the impact on the binding of specific shelterins is likely to differ between these two hTR mutants, 47A and TSQ hTR induced similar extents of TIFs (comparing between either WT-TRF1HP1 α expressing cells, or Vonly cells; Fig. 4b and c).

I165A- and W174A-TRF1HP1 α alleles were both unable to inhibit telomerase action (Fig. 3f-i), and also showed more mutant 47A hTR-induced TIFS than WT-TRF1HP1 α (Fig.4a-b). From this finding alone, it was not possible to determine if this increased telomere damage (more TIFs) was the result of more incorporation of 47A repeats on telomeres than with WT-TRF1HP1 α , or whether the telomeres were less protected via a different mechanism. Therefore, in order to distinguish whether any HP1 α allele-specific effect on protection from mutant hTR-induced telomere damage was caused by its having an effect on telomere protection (assessed by protection from TIFs), as opposed solely to different extents of incorporation of the mutant DNA sequence by telomerase, we only compared WT TRF1HP1 α specifically with those of its

mutants (V22M, NS2A and KRKAAA) that showed similar WT hTR incorporation efficiencies (Fig 3f-i).

We agree with the reviewer that indeed we cannot exclude the possibility that the incorporation efficiency by telomerase of a given mutant hTR may differ compared to WT hTR. However, because all four fusion HP1 α constructs compared - WT TRF1HP1 α , V22M, NS2A and KRKAAA constructs - inhibited incorporation upon WT hTR over-expression to the same extent, we would expect that, for any specific mutant hTR sequence, WT TRF1HP1 α , V22M, NS2A and KRKAAA would have a similar effect on incorporation driven by telomerase action.

In other words, even though incorporation of WT hTR may be different from mutant hTR, so long as we are comparing within the same mutant hTR sequence across these HP1 α allele groups, our interpretation should still be valid.

In addition, to directly study the protective effect of TRF1HP1 α independent of hTR, we have now included new experimental data (Fig 5) by inducing telomeric-specific damage via TRF2 depletion, as described in details below in responding to the following suggestion by the reviewer.

In order to clearly show the telomere protective effect of HP1, other telomere dysfunction setting (such as TRF2 or POT1 downregulation) should be studied.

RESPONSE: We agree, and thank the reviewer for this suggestion. Our initial submission manuscript did not find clear evidence for a telomere protective effect of the tethered fusion protein based on the criteria of TIFs, senescence (Fig. 2d-h) or cell growth rates (Supplementary Fig. 5) but rather, the effect of the tethered protein we found is on telomerase action on telomeres (Fig. 2b-c; Fig. 3f-i), which is a limited and more specific aspect of what can be considered as telomere “protection”. In addition, we identified two potential separation of function mutants (NS2A and KRKAAA) (Fig. 4a-b) that reduced the manifestation of telomere damage (TIFs) resulting from incorporated mutant telomeric DNA repeats. The NS2A and KRKAAA findings indicated that TRF1-tethered HP1 α wither protects telomeres from being damaged, and / or plays an active role in DNA damage repair at the telomeres.

We now have added two new figures that directly study the protective effect of HP1 α :

- 1) TRF2 experiment (Fig. 5): WT-TRF1HP1 α mildly protects from telomere damage (TIFs) induced upon efficient TRF2 knockdown (Fig. 5a, c) compared to the non-targeting si-RNA controls (Fig. 5a, b). Furthermore, comparing the pattern across all of the TRF1HP1 α mutants, each of the allele-specific effects on TRF2-depletion-induced TIFs (Fig. 5c) closely paralleled their corresponding effects on 47A-hTR- induced TIFs (Figure 4b). The 47A mutated DNA sequence is known to prevent high affinity binding of TRF2. This similarity of protective effects against both telomerase-independent (TRF2 knock-down) and 47A hTR-induced damage indicates that telomere-tethered WT-TRF1HP1 α can additionally protect telomeres independently of its inhibitory effect on telomerase action.

Independently, we also used

- 2) A new CRISPR-based approach to specifically induce telomere-specific damage (Supplementary Fig. 6). Interestingly, over-expressed TRF1 alone is sufficient to protect cutting of telomeres by CRISPR. WT-TRF1HP1 α was comparably protective.

Thus, by employing independent approaches to induce telomeric damage, we learn different aspects about how HP1 α tethering affects telomere protection.

- What is the level of HP1 naturally present at telomeres in the cell line studied
How much it is increased upon TRF1-HP1?

RESPONSE: In UM-UC3 cell line, we detected low endogenous HP1 α at telomeres in ChIP immunoprecipitated with HP1 α antibody (Fig 1f-g), consistent with the previously reported very low natural levels of HP1 α at human telomeres⁹⁻¹³. ChIP shows ~21 fold increase of telomere-tethered TRF1HP1 α at the telomeres (Fig 1f-g).

What happens if the expression of the endogenous HP1 gene is downregulated?

RESPONSE: We thank the reviewer of pointing out this approach that we have considered but did not perform such experiments for the following two reasons:

1) Consistent with what others previously reported^{9, 11-13}, we also only observed low levels of HP1 α at human telomeres in UM-UC3 (Fig 1f-g). A recent NAR publication¹⁰ also reported low H3K9me3 at telomeres in multiple human cell lines. Thus, we expect downregulating endogenous HP1 α to only yield minimal effect (if any) on telomere maintenance in our setup. Therefore, by experimentally tethering HP1 α to telomeres, our approach was chosen to produce clear signal-to-noise results. 2) Moreover, should any alterations of telomere maintenance be observed upon HP1 α knock-down, they would likely result from indirect effects on gene expression from genome wide HP1 α reduction, and thus complicate interpretation.

- The effect of the HP1 mutants impairing telomere elongation should be studied in a cell line engineered to express the mutants from the endogenous HP1 locus

RESPONSE: We thank the reviewer for the suggestion. However, we believe this would complicate the analysis as we would be replacing normal HP1 α , and HP1 α genome wide effects would potentially occur that were not specific to telomeres. As a result, it would be challenging to isolate effects specific to telomeres. We would therefore lose the power of our tethering strategy. In addition, we respectfully point out to the reviewer that we have carefully included various negative controls (Vonly, TRF1 alone or HP1 α alone) (Fig. 2), together with point mutants within HP1 α of the TRF1HP1 α (Fig. 3). Therefore, we do not believe that our results are the consequence of overexpression, as explained in detail above. Thus, we feel that engineering a new cell line is beyond the scope of this study.

Minor

- The increment in H3K9me3 upon TRF1HP1 overexpression is very little and not really visible from the slotblot of figure 1f.

RESPONSE: We reported an increment in H3K9me3 per H3 (ie. normalized to H3). We have now also included replicates of three independent ChIP dot blots raw results (Supplementary Fig 1) showing a consistent, slightly observable decrease of H3 in TRF1HP1 α (thus an increment of H3K9me3 per H3) compared to controls.

For the reviewer only: In addition, we consistently observed decreased H3 normalized to 10% input (a, b) and increased H3K9me3 per H3 (a, c) in the context of hTR overexpression as well.

Reviewer #3 (Remarks to the Author):

In Chow et al. the authors describe a novel use of TRF1 as tethering tool to enrich specific proteins at telomeres, in this case HP1a WT and mutant variants. Through this approach, the authors show that HP1a enrichment at telomeres can counteract telomere lengthening in hTR over-expression conditions and can induce alterations in telomere structure. The findings described are interesting for the field of telomeres and HP1 function. In particular, the use of HP1a point mutants allows to uncouple different activities of HP1a, highlighting that H3K9me3 binding is dispensable for the observed phenotypes and suggesting that the dimerization of HP1a and the binding of an unknown ligand are involved. Although these observations are exciting the work scratches the surface of the molecular events behind them. Further characterisation of the HP1a ligands that prevent telomere lengthening and mediate telomere reorganisation would greatly reinforce the message of this work. Beside this aspect that could be beyond the scope of this work, additional points should be addressed to fully support the conclusions proposed.

RESPONSE: We are glad the reviewer thinks this work will be of interest to the telomere and HP1 field. We agree with the reviewer that the results are exciting and feel that communicating these first sets of new observations is essential before we, and others in the field can assess in detail over the next several years how HP1alpha ligands regulate the phenotypes.

Major points:

- Most importantly, the authors don't discern in a clear way the effects related to TRF1-HP1a tethering to telomeres to the ones related to TRF1 overexpression itself. Indeed, the overexpression of TRF1 alone causes a slight decrease of H3 compared to Vonly (ChIP, Fig 1h) and an increase in Rg and average number of localisations per telomere compared to HP1a (STORM, Fig 5h, line 260).

RESPONSE: We thank the reviewer for raising a good point that we can clarify.

For ChIP, the slight decrease of H3 in TRF1 compared to Vonly ($p=0.078$; $n=3$) or HP1α ($p=0.1677$; $n=3$) is not significant - assessed by one-way ANOVA and Dunnett's multiple comparison test with 95% confidence level. Importantly, this slight decrease does not affect the

functional readouts of telomere elongation (Fig. 2b-c) and TIF formation (Fig. 4a-c), which form a central message in our working model, clearly indicating that HP1 α tethered at telomeres by TRF1 (i.e. TRF1HP1 α) inhibits telomerase action but not TRF1 alone, HP1 α alone nor Vonly (Fig. 2b-c). We have now more clearly stated these controls in the main text for Fig 2b-c (lines 173-176). Furthermore, our new experimental data (Fig. 5) - added to this revised manuscript - now shows that in addition, in HP1 α allele-specific fashion, telomere-tethered HP1 α protects telomeres against telomerase-independent telomere damage.

For the STORM Rg, as noted in the text, the different amount of Rg increase in TRF1 overexpression versus HP1 α overexpression is indeed an interesting phenotype. We are studying this phenomenon as a separate project, while in this manuscript, we focus on TRF1HP1 α having a higher Rg than either TRF1 or HP1 α . In fact, our single point mutation TRF1HP1 α 165A, together with TRF1, made an incisive and compelling pair of negative controls. TRF1HP1 α 165A resulted in a decreased Rg compared to TRF1HP1 α , similar to that of TRF1 in old Fig. 5h-i (now renumbered as Fig. 6h-i).

Regarding the increased average number of localization points in TRF1 compared to HP1 α alluded to by the reviewer, we respectfully point out that the number of localization points is primarily used for quality controls during telomere identification (>200 localization points). The localization point numbers depend not only on the heterogeneity of telomere lengths within a nucleus, but also on labeling efficiency (e.g. probe accessibility) and experimental conditions (affecting the blinking kinetics of the fluorophores), whereas Rg is more strictly dependent on the shape of the telomere. In fact, to address this point, scatter plots of Rg versus localization point number for individual telomeres across several experimental conditions show only weak correlation (Supplementary Fig. 7 and Supplementary Fig. 8d,e). Hence, the difference in the average number of localization points should not drastically alter the statistics of Rg. We have updated our text (lines 301-310) and added Supplementary Fig. 7 and Supplementary Fig. 8d-e to clarify this point.

Also, “long term culturing conditions of TRF1 overexpression in certain cancers resulted in telomere shortening³⁹” (line 161). Given this, Vonly controls should be included in STORM analyses and every experimental group (Vonly, TRF1 and HP1 α) should be considered separately to compute statistics in CHIP experiments.

RESPONSE: We respectfully point out to the reviewer that for STORM, we did state (lines 288-290) that we “first verified that all experimental groups, collected at earliest passage after blasticidin selection (Day 8 to 9 post lentiviral infection)”, and showed our cell populations had similar telomere lengths across the different experimental groups tested (Fig. 6a). We have now additionally included Vonly and parental controls indicating no significant difference in average Rg(nm) (Supplementary Fig. 8a-c).

CHIP was performed in samples with similar telomere length across groups (Supplementary Fig. 3) and at the earliest time points possible. As detailed above (i.e. response to the reviewer’s first comment), in Fig 1h, the slight decrease in TRF1 per H3 compared to Vonly ($p=0.078$; $n=3$) or HP1 α ($p=0.1677$; $n=3$) are not statistically different. Moreover, all controls (Vonly, TRF1 and HP1 α) showed a consistent pattern of increased H3 compared to TRF1HP1 α . Thus, we feel it is more appropriate to compute the statistic by grouping all negative controls together.

- Although TRF1-HP1 α tethers HP1 α to telomeres, TRF1 will in turn be tethered to HP1 α

binding sites. The use of mutants only partially addresses this point. The authors should discuss the potential implications of the aberrant delivery of TRF1 to non telomeric regions.

RESPONSE: We agree that this could have been of concern. Fortunately, the design and results of our mutational studies (Fig. 3b-c), along with TRF1 alone controls (Fig. 1b-c), did successfully and effectively address the potential of TRF1 delivery to non-telomeric regions. Mutant V22M was deliberately chosen to control for any indirect effects due to aberrant delivery of TRF1 (or excess HP1 α) to other genomic regions. This is in part because V22M has lost its ability to be recruited via H3K9me3 at non-telomeric regions. V22M single and double mutants (V22MI165A) were exclusively tethered by TRF1 at the telomeres, and not to other genomic regions (Fig. 3b-d). Importantly, at least within our system, V22M and WT TRF1HP1 α both limited telomerase extension to similar and hence only differed minimally, at most, in their effect on telomerase action (Fig 3b-d, Fig 3f, h). Similarly, the phenotype of double mutant (V22MI165A) could not be accounted for by TRF1 going to other genomic regions (Fig 3b-d, Fig 3f, h). In addition, we also included additional controls with either TRF1 alone or HP1a alone (Fig 2b-c). Thus, telomerase inhibition by TRF1HP1a is unlikely to be confounded by indirect effects. We realize the rationale description for these experiments was not clear in our originally submitted manuscript, and have more clearly stated the reasons and results of these control experiments within the revised main text (lines 197-213).

Do the authors envisage technological improvements to overcome this limitation?

RESPONSE: We considered using dCas9 CRISPR to tether HP1a at telomeres. However, CRISPR-dCas9 is much bulkier than TRF1, and lacks the normal interactions with other shelterin components and telomeric DNA provided by the tethering TRF1 molecules. It is also harder to predict how CRISPR helicase activity etc. will impact telomere integrity. TRF1 tethering, on the other hand, is relatively more natural in preserving telomere integrity.

- Colocalisation analyses: how does colocalisation is measured in this study? A methodological description should be added in the Methods section.

RESPONSE: Quantification of co-localization has been elaborated in the "Method" section (lines 564-568).

Also, the extent of colocalisation between TRF2 and HP1 is very high. How the authors explain this observation? A short sentence is included in the figure legend but it is hard to verify in the images.

RESPONSE: We thank the reviewer for the opportunity to clarify. We have updated the associated legend of Fig 1d: The high apparent colocalization of HP1 α with TRF2 (within the HP1 α group) is partly caused by random, co-incidental overlaps with telomeres due to widespread HP1 α spots; X-Y planes are projections of z-stacks.

To this regard, bigger images or zoom in panels would help the interpretation of results.

RESPONSE: We have now included zoomed-in panels within Fig 1b, 3b, 4a and 4d.

- ChIP experiments: in the introduction, the authors mention that knockout of SUV39H1/2 and SUV4-20H1/2 result in defective telomere function and telomere lengthening. It would be interesting to verify by ChIP if H4K20me3 levels and/or SUV4-20H recruitment is affected by TRF1-HP1a.

RESPONSE: This is a good point of the reviewer, and indeed we have looked into H4K20me3 levels to understand the possible effect on H4K20me3 upon introduction of TRF1HP1a.

For reviewer only: **(a-b)** We indeed saw decreased H3 at the telomeres (normalized to 10% input) while increased H4K20me3 per H3 at the telomeres **(a, d)**, in a similar pattern as H3K9me3 **(a, c)**. However, the H4K20me3 signal was much weaker than H3K9me3 **(a, c, d)**. We chose to focus on H3K9me3 as it is most relevant to HP1a. The robust TRF2 signal showed telomeric probe specificity **(a)**. In contrast, as expected there was minimal H3K9Ac **(a, e)**.

Indeed, direct binding between HP1a has been described (Schotta G et al Genes & Dev 2004; Hahn M et al Genes & Dev 2013) and the binding contributes to chromatin compaction at centromeres through recruitment of Cohesin (Hahn M et al Genes & Dev 2013). Also, SUV4-20H deficiency results in telomere lengthening in MEFs and mESCs (Benetti R et al JCB 2007). The involvement of SUV4-20H in the phenotype observed would be an intriguing possibility and observing an increase in H4K20me3 would reinforce the concept of heterochromatin mark accumulation shown by a slight increase in H3K9me3.

RESPONSE: We appreciate the reviewer's comments about reconciling our results with published work. As described above, we performed ChIP with H4K20me3. Upon TRF1HP1a expression, we observed a similar pattern of enrichment at telomeres as in H3K9me3. Here for the reviewer only - the requested data is shown above. As expected, on the contrary, there was

minimal H3K9Ac. However, since the H4K20me3 signal was much weaker than H3K9me3, we chose to focus on H3K9me3 because it is most relevant to HP1 α .

- STORM experiments: the authors calculate the Rg of telomeres to characterise alterations in telomere shape. This type of analysis requires very accurate drift correction as suboptimal drift correction can lead to deformation of telomeres in super-resolution. Also, the low number localisation/frame typically found in this type of imaging, can make drift correction challenging. Do the authors use beads? If this is the case it should be mentioned. If beads were not used, authors should confirm their observations in the presence of beads.

RESPONSE: We appreciate the concerns regarding the drift correction and thank the reviewer for the opportunity to make clear how we address this important issue. We completely agree that drift correction is critical for super-resolution microscopy. Therefore, drift correction has been always carefully performed in our STORM analysis. We did not use beads as fiducial markers because beads on the coverslip are on a different focal plane than the telomeres inside the nuclei. Instead, to address this important issue, we used two independent methods for drift correction.

For reviewer only: The first method (a-c) is based on image correlation of bright-field (BF) images

of the cell, which we record once every five frames during STORM data acquisition. The high-signal bright field images turn the cells themselves into fiducial markers¹⁴. The second method is based on correlation analysis of the STORM localization points¹⁵. Because there are usually 30 detected telomeres per nuclei, there are actually a sufficient number of localization points to extract the sample drift during acquisition. To prove this point, we compared the Fourier Ring Correlation (FRC) of “Parental” and “Vonly”

images drift-corrected using either method (d-e). Due to the repetitive switching of dyes, FRC in this case will mostly reflect the combined effect from localization uncertainty and residual drift¹⁵. There was little difference between FRC values using these two drift correction methods. We note that FRC value is by nature always larger than the localization precision (~ 30 nm FWHM in the xy plane in our case). Moreover, a 2D Gaussian distribution with a FWHM of 30 nm has an Rg of $30/2.35 = 13$ nm (even a FWHM of 60 nm gives an Rg of just 26 nm), whereas the smallest telomere Rg we have is 35 nm. Thus, the telomere Rg calculations in this manuscript are fully

supported by the resolution of our images. Therefore, for all subsequent STORM data reported in the manuscript, we used the localization points based method for its simplicity.

Also, how telomeres are identified and analysed?

Are they manually selected? Are they automatically identified with clustering algorithms? Please provide this information in the methods section.

RESPONSE: The individual telomeres were selected manually. We have added the selection methods and the criteria in the methods section (lines 610-613).

Since TRF1 and HP1a controls show different Rg values and average number of localisations, a Vonly control should be introduced to determine the Rg in a more neutral control condition and to define whether TRF1 overexpression alters telomere structure.

RESPONSE: We thank the reviewer for the comments. As mentioned above in our response to the first set of comments by the reviewer, we have now included a new figure (Supplementary Fig. 8) showing Vonly and parental results in similar mean Rg (nm), and we have updated the text accordingly (lines 308-310). We also respectfully point out that the number of localization points is primarily used for quality controls during telomere identification (>200 localization points). The localization point number depends not only on the heterogeneity of telomere lengths across chromosomes, but also on labeling efficiency (e.g. probe accessibility) and experimental conditions (affecting the blinking kinetics of the fluorophores), whereas Rg is more strictly dependent only on the shape of the telomere. In fact, we added scatter plots of Rg versus localization point number for individual telomeres (new Supplementary Fig. 7 and Supplementary Fig. 8d-e), showing only weak correlation for all experimental groups.

All changes have been made in the revised manuscript to comply with the reviewer's suggestions below.

Minor points:

- The authors should clearly cite the references referring to each of the mutants described in the manuscript and include in the methods section the source of the constructs or how they were generated. Done.

- Telomere length analyses in Figure 2b and especially in Supplementary Figure 2b are not clearly visible. Higher exposure images should be shown.

A higher exposure image has been added in Old Supplementary Fig. 2b (now renumbered as Supplementary Fig. 3)

- Adding colours to Supplementary Figure 3b would help the visual interpretation of the graph.

Colors are added to Old Supplementary Fig. 3b (now renumbered as Supplementary Fig. 5).

- Line 257: "we only analysed telomere clusters with centers of mass near the focal plane, and consisting of more than 200 localization points (Fig. 5c)." Figure 5c does not refer to this.

We regret the confusion due to our wording, which we can clarify. Old Fig. 5c (now renumbered as Fig 6c) shows representative examples of the telomeres that we analyzed, which as described in the text, only include those with over 200 localization points (lines 303-305). Accordingly, we have re-written the figure legend to clarify this panel: The numbers above the images in Fig 6c refer to the particular measured Rg (nm) for the telomere shown in each case as two images: Top row green image is the reconstructed image across a gradient of Rg; Bottom row is the raw image as acquired (all with >200 localization points) prior to image processing and reconstruction to create the cleaned up green image above.

- Line 259: "The average number of localization points of such filtered individual telomeres for TRF1, HP1 α , WT TRF1HP1 α , TRF1HP1 α I165A were 664, 420, 542, and 640, respectively." Please add standard deviation values to include information about the variability of localisations detected.

We have indicated standard deviation accordingly within the figure legend of (Supplementary Figs. 7 and 8). These figures also show the distribution of localization points corresponding to its Rg (nm).

- The authors should include a section describing the statistical methods applied in the methods section.

We have added a paragraph describing statistical analyses in the methods section (lines 615-621).

- TRF1-HP1a and senescence: the authors show that TRF1-HP1a does not exacerbate the levels of B-Gal positive cells in high passage BJ and WI38 cells. Could TRF1-HP1a instead anticipate the appearance of B-Gal positive cells when overexpressed in low passage cells?

We thank the reviewer for this excellent point that we can address. We reasoned based on considerable previous reports by others, and our own observations previously, if there was an effect on senescence we likely would get the best readout signals by using cells that were poised to undergo experimentally observable senescence. Higher passage cells generally are more sensitive to exogenous perturbation compared to lower passage cells. Besides beta gal staining intensity, another determinant of the onset of senescence is the induction of TIFs. We observed similar, minimal baseline levels of TIFs comparing TRF1HP1 α and its controls (Fig. 4f) at the earliest time-points possible, when telomere length has undergone no or only differences across the experimental groups. Therefore, our combined results show that the fusion protein does not make the cells sicker. Performing a beta gal experiment on younger passage cells is not likely to provide additional information to our working model. While we feel this experiment falls outside the scope of the focus of our working model, this is interesting for future investigation in other studies.

- Lines 549 and 550: the sentence does not refer to STORM imaging, please check and move to correct section.

We have now placed this back in its appropriate within the ChIP method section.

- In the Discussion the main biological impact of their findings should be stressed out. Also the discussion about TRF2 might be reduced or eliminated, considering the results presented.

We feel the discussion about TRF2 conveys an important message, relevant to our interpreting our experimental results; regarding how telomere chromatin impacts its maintenance, it is important to take into consideration how our experimental set-up may influence other core shelterin components. Thus, we respectfully have decided to keep the TRF2 discussion.

With respect to the biological impact, we have elaborated a paragraph below in the discussion (line 427-440) highlighting the implication of our findings.

REFERENCES

1. Canzio, D., Larson, A. & Narlikar, G.J. Mechanisms of functional promiscuity by HP1 proteins. *Trends Cell Biol* **24**, 377-386 (2014).
2. Oliva-Rico, D. & Herrera, L.A. Regulated expression of the lncRNA TERRA and its impact on telomere biology. *Mech Ageing Dev* **167**, 16-23 (2017).
3. Arnoult, N., Van Beneden, A. & Decottignies, A. Telomere length regulates TERRA levels through increased trimethylation of telomeric H3K9 and HP1alpha. *Nat Struct Mol Biol* **19**, 948-956 (2012).
4. Farnung, B.O., Brun, C.M., Arora, R., Lorenzi, L.E. & Azzalin, C.M. Telomerase efficiently elongates highly transcribing telomeres in human cancer cells. *PLoS One* **7**, e35714 (2012).
5. Smirnova, A. *et al.* TERRA Expression Levels Do Not Correlate with Telomere Length and Radiation Sensitivity in Human Cancer Cell Lines. *Front Oncol* **3**, 115 (2013).
6. Van Beneden, A., Arnoult, N. & Decottignies, A. Telomeric RNA expression: length matters. *Front Oncol* **3**, 178 (2013).
7. Martinez, P. *et al.* Increased telomere fragility and fusions resulting from TRF1 deficiency lead to degenerative pathologies and increased cancer in mice. *Genes Dev* **23**, 2060-2075 (2009).
8. Sfeir, A. & de Lange, T. Removal of shelterin reveals the telomere end-protection problem. *Science* **336**, 593-597 (2012).
9. Rosenfeld, J.A. *et al.* Determination of enriched histone modifications in non-genic portions of the human genome. *BMC Genomics* **10**, 143 (2009).
10. Cubiles, M.D. *et al.* Epigenetic features of human telomeres. *Nucleic Acids Res* **46**, 2347-2355 (2018).
11. Minc, E., Allory, Y., Worman, H.J., Courvalin, J.C. & Buendia, B. Localization and phosphorylation of HP1 proteins during the cell cycle in mammalian cells. *Chromosoma* **108**, 220-234 (1999).
12. O'Sullivan, R.J., Kubicek, S., Schreiber, S.L. & Karlseder, J. Reduced histone biosynthesis and chromatin changes arising from a damage signal at telomeres. *Nat Struct Mol Biol* **17**, 1218-1225 (2010).
13. Sharma, G.G. *et al.* Human heterochromatin protein 1 isoforms HP1(Hsalpha) and HP1(Hsbeta) interfere with hTERT-telomere interactions and correlate with changes in cell growth and response to ionizing radiation. *Mol Cell Biol* **23**, 8363-8376 (2003).
14. McGorty, R., Kamiyama, D. & Huang, B. Active Microscope Stabilization in Three Dimensions Using Image Correlation. *Opt Nanoscopy* **2** (2013).
15. Wang, Y. *et al.* Localization events-based sample drift correction for localization microscopy with redundant cross-correlation algorithm. *Opt Express* **22**, 15982-15991 (2014).

REVIEWERS' COMMENTS:

Reviewer #1 (Remarks to the Author):

The authors satisfactory addressed my comments. They also clarified all points, as requested, increasing the comprehension of the manuscript

Reviewer #2 (Remarks to the Author):

In the revised version the authors satisfactory addressed my points.

Reviewer #3 (Remarks to the Author):

The authors have appropriately addressed the points raised by reviewers with new experimental data supporting their conclusions. Unclear aspects of the manuscript have now been clarified. There are no further concerns.